# Large-scale characterization of drug mechanism of action using proteome-wide thermal shift assays

Jonathan G Van Vranken[1], Jiaming Li[1], Julian Mintseris[1], Ting-Yu Wei[1], Catherine M Sniezek[2], Meagan Gadzuk-Shea[2], Steven P Gygi[1]*, Devin K Schweppe[2]*

[1]Department of Cell Biology, Harvard Medical School, Boston, United States; [2]Department of Genome Sciences, University of Washington, Seattle, United States

## eLife Assessment

The study provides a **valuable** showcase of a workflow to perform large-scale characterization of drug mechanisms of action using proteomics in which on-target and off-targets of 166 compounds using proteome solubility analysis in living cells and cell lysates were determined. The evidence supporting the claims of the authors is **solid**, however, the inclusion of more replicate experiments and more statistical rigor would have strengthened the study. This will be of broad interest to medicinal chemists, toxicologists, computational biologists and biochemists.

**\*For correspondence:**
steven_gygi@hms.harvard.edu (SPG);
dkschwep@uw.edu (DKS)

**Abstract** In response to an ever-increasing demand of new small molecules therapeutics, numerous chemical and genetic tools have been developed to interrogate compound mechanism of action. Owing to its ability to approximate compound-dependent changes in thermal stability, the proteome-wide thermal shift assay has emerged as a powerful tool in this arsenal. The most recent iterations have drastically improved the overall efficiency of these assays, providing an opportunity to screen compounds at a previously unprecedented rate. Taking advantage of this advance, we quantified more than one million thermal stability measurements in response to multiple classes of therapeutic and tool compounds (96 compounds in living cells and 70 compounds in lysates). When interrogating the dataset as a whole, approximately 80% of compounds (with quantifiable targets) caused a significant change in the thermal stability of an annotated target. There was also a wealth of evidence portending off-target engagement despite the extensive use of the compounds in the laboratory and/or clinic. Finally, the combined application of cell- and lysate-based assays, aided in the classification of primary (direct ligand binding) and secondary (indirect) changes in thermal stability. Overall, this study highlights the value of these assays in the drug development process by affording an unbiased and reliable assessment of compound mechanism of action.

## Introduction

A critical aspect of any drug development effort is the ability to define compound mechanism of action. Most small molecule therapeutics function by engaging a protein (target) through a direct physical interaction, thereby inhibiting or activating a specific target in order to modulate enzymatic activity, cell signaling pathways, or gene expression. Because of its unique ability to probe cell function on a proteome-wide scale, mass spectrometry (MS) has emerged as a powerful tool for interrogating compound mechanism of action by aiding in the validation of known targets and the identification of new ones (*Moellering and Cravatt, 2012*; *Daub, 2015*; *Schürmann et al., 2016*; *Frantzi et al.,*

*2019*; *Robers et al., 2020*; *Kuljanin et al., 2021*; *Meissner et al., 2022*). In addition to traditional proteomic approaches that profile protein expression and post-translational modifications (PTMs), the mass spectrometer has also become indispensable for unbiased assessments of compound target engagement and, by extension, mechanism of action. This has led to the development of numerous MS-based methods to screen the proteome for evidence of ligand binding (*Lomenick et al., 2009*; *Molina et al., 2013*; *Jafari et al., 2014*; *Savitski et al., 2014*; *Liu et al., 2017*; *Gaetani et al., 2019*; *Ball et al., 2020*; *Van Vranken et al., 2021*; *Beusch et al., 2022*).

All proteins have a characteristic melting temperature, and this physical property can be influenced by numerous factors (ligand-binding, protein-protein interactions, and PTMs) that make a given protein more or less resistant to thermal denaturation (*Koshland, 1958*). Classically, the thermal shift assay was used to study a single protein of interest, however, by coupling the traditional thermal shift assay with modern quantitative proteomics and sample multiplexing, it is possible to assess changes in protein thermal stability on a proteome-wide scale (*Molina et al., 2013*; *Jafari et al., 2014*; *Savitski et al., 2014*). As such, MS-coupled thermal shift assays have emerged as a powerful tool for studying compound target engagement and mechanism of action in lysates, cells, and even tissues (*Molina et al., 2013*; *Savitski et al., 2014*; *Jafari et al., 2014*; *Franken et al., 2015*; *Huber et al., 2015*; *Reinhard et al., 2015*; *Becher et al., 2016*; *Mateus et al., 2016*; *Sridharan et al., 2019a*; *Mateus et al., 2020*; *Perrin et al., 2020*).

The cellular thermal shift assay (CETSA) and thermal proteome profiling (TPP) use tandem mass tag (TMT)-based quantitative proteomics to quantify complete melting curves and assign melting temperatures ($T_M$) to thousands of proteins, simultaneously, making it possible to screen the proteome for ligand-induced changes in protein thermal stability (*Molina et al., 2013*; *Savitski et al., 2014*; *Jafari et al., 2014*; *Perrin et al., 2020*; *Zinn et al., 2021*). More recently, the proteome integral solubility alteration (PISA) assay was proposed to improve the efficiency of the proteome-wide thermal shift assay (*Gaetani et al., 2019*). Rather than building complete melting curves, PISA uses TMT-based quantitative proteomics to estimate, or integrate, the area under a protein melting curve (*Figure 1—figure supplement 1A*). Instead of TMT labelling the soluble fraction from each individual temperature—as in CETSA and TPP—PISA pools the soluble fractions of multiple samples heated across a thermal gradient such that a single TMT reporter represents an entire integrated melting curve. So, while CETSA and TPP measure a change in melting temperature ($\Delta T_M$), PISA measures a change in solubility ($\Delta S_M$). Critically, there is a strong correlation between $\Delta T_M$ and $\Delta S_M$, which makes PISA a reliable, if still imperfect, surrogate for measuring direct changes in protein thermal stability (*Gaetani et al., 2019*; *Li et al., 2020*). Thus, in the context of PISA, a change in protein thermal stability (or a thermal shift) can be defined as a fold change in the abundance of soluble protein in a compound-treated sample vs. a vehicle-treated control after thermal denaturation and high-speed centrifugation. Therefore, an increase in melting temperature, which one could determine using CETSA or TPP, will lead to an increase in the area under the curve and an increase in the soluble protein abundance relative to controls (positive $\log_2$ fold change). Conversely, a decrease in melting temperature will result in a decrease in the area under the curve and a decrease in the soluble protein abundance relative to controls (negative $\log_2$ fold change). In the end, the ability to compress entire melting curves into a single TMT channel enables the synchronized interrogation of multiple compounds at multiple concentrations with multiple replicates in one experiment (*Gaetani et al., 2019*).

Owing to improved scalability, PISA results in an eightfold theoretical improvement in throughput compared to TPP and CETSA. Therefore, PISA makes it feasible to readily approach large-scale chemical perturbation experiments, such as chemical library screening. With this in mind, we sought to establish an efficient platform for screening chemical libraries for drug-induced changes in protein thermal stability and test the ability of this workflow to provide meaningful compound-specific data when performed at scale. We selected 96 compounds with known mechanisms of action for PISA screening in live K562 cells and further assayed 70 of the compounds in native lysates. Using these data, we identified evidence of on-target binding, as well as putative examples of off-target engagement. In addition to direct drug binding, we were able to identify secondary changes in protein thermal stability that did not result from direct engagement and, instead, likely stemmed from other factors such as PTMs or changes in macromolecular interactions. Overall, this study serves as a roadmap for the scalable implementation of protein thermal stability quantitation to interrogate compound mechanism of action.

## Results

### Establishing a workflow for large-scale chemical perturbation screening

To enable large-scale chemical perturbation screening, we first sought to establish a robust workflow for assessing protein thermal stability changes in living cells. We chose K562 cells, which grow in suspension, because they have been frequently used in similar studies and can easily be transferred from a culture flask to PCR tubes for thermal melting (*Savitski et al., 2014*; *Jarzab et al., 2020*). In a PISA experiment, a change in melting temperature or a thermal shift is approximated as a significant deviation in soluble protein abundance following thermal melting and high-speed centrifugation. Throughout this manuscript, we will interpret these observed alterations in solubility as changes in protein thermal stability. Most commonly this is manifested as a $\log_2$ fold change comparing the soluble protein abundance of a compound-treated sample to a vehicle-treated control (*Figure 1—figure supplement 1A*). The ranges of temperatures used in a PISA experiment can impact the magnitude of the fold-change measurements, with narrower temperature ranges often resulting in larger fold changes (*Li et al., 2020*). After testing a number of ranges experimentally, we ultimately settled on a temperature range of 48–58°C, which encompasses the back half of most protein melting curves in K562 cells (*Figure 1—figure supplement 1B*; *Jarzab et al., 2020*). Importantly, similar temperature ranges have been utilized in other studies that employ PISA for target deconvolution (*Sabatier et al., 2022*).

To highlight the advantage of this window, K562 cells were treated with each of the clinically available CDK4/6 inhibitors—ribociclib, abemaciclib, and palbociclib—or DMSO (10 µM, 30 min; *Figure 1—source data 1*). Notably, all three compounds and a DMSO control were able to be assayed with four biological replicates in a single TMTPro 16plex experiment, requiring just 24 hr of instrument time. The cells were either heated across a range of 37–62°C (to approximate a full melting curve) or 48–58°C (thermal window) and any changes in protein thermal stability (based on a deviation in soluble protein abundance after thermal denaturation) were assessed using PISA. In cells heated across the full melting curve, CDK4 and CDK6 experienced minimal $\log_2$ fold changes, whereas utilizing the narrower range yielded a statistically significant change in CDK4 and CDK6 abundance following thermal denaturation and centrifugation, which, in a PISA experiment, is consistent with a ligand-induced thermal shift or, more specifically, a thermal stabilization (*Figure 1—figure supplement 1C*). In addition to CDK4/6, several other kinases experienced a significant change in solubility (thermal stability) in response to treatment with each compound (*Figure 1—figure supplement 1D–G*). In fact, each of the three CDK4/6 inhibitors appear to engage a unique set of kinases, which might contribute to their disparate clinical and molecular phenotypes (*Figure 1—figure supplement 1G*; *Hafner et al., 2019*). Finally, we observed a consistent inhibitor-induced negative $\log_2$ fold change in soluble protein abundance for RB1, a well-established phosphorylation target of CDK4/6, which is consistent with a thermal destabilization (*Figure 1—figure supplement 1C–F*; *Narasimha et al., 2014*). The simplest explanation of this change is that inhibition of CDK4/6 prevents phosphorylation of RB1 thereby inducing a change in its thermal stability.

### A large-scale chemical perturbation screen in K562 cells

We next sought to execute a large-scale chemical screen in live K562 cells. Specifically, we hoped to test the ability of PISA to provide meaningful insights into compound mechanism of action when performed at scale in the context of a chemical library screen. To that end, we curated a custom chemical library comprised of 96 commonly used cancer drugs and tool compounds. Each compound selected for the library had at least one well-annotated target and a known mechanism of action (*Supplementary file 1*). Of these 96 compounds, 70 targeted protein kinases, while the remaining 26 compounds targeted other classes of proteins including histone deacetylases (HDACs), lysine demethylases, poly (ADP-ribose) polymerases (PARPs), and others (*Figure 1A*). The library also contained multiple compounds targeting certain proteins—eight HDAC inhibitors, four aurora kinase inhibitors, six BRAF inhibitors, etc. To initiate the screen, K562 cells were treated with 10 µM of each compound for 30 min in biological duplicate, which is consistent with previous thermal proteome profiling work (*Figure 1A*; *Savitski et al., 2018*; *Herneisen and Lourido, 2021*; *Johnson et al., 2023*). The live cells were heated across a thermal gradient (48–58°C) and any changes in protein thermal stability were approximated using PISA (*Figure 1—figure supplement 2*). This resulted in a total of 256 samples,

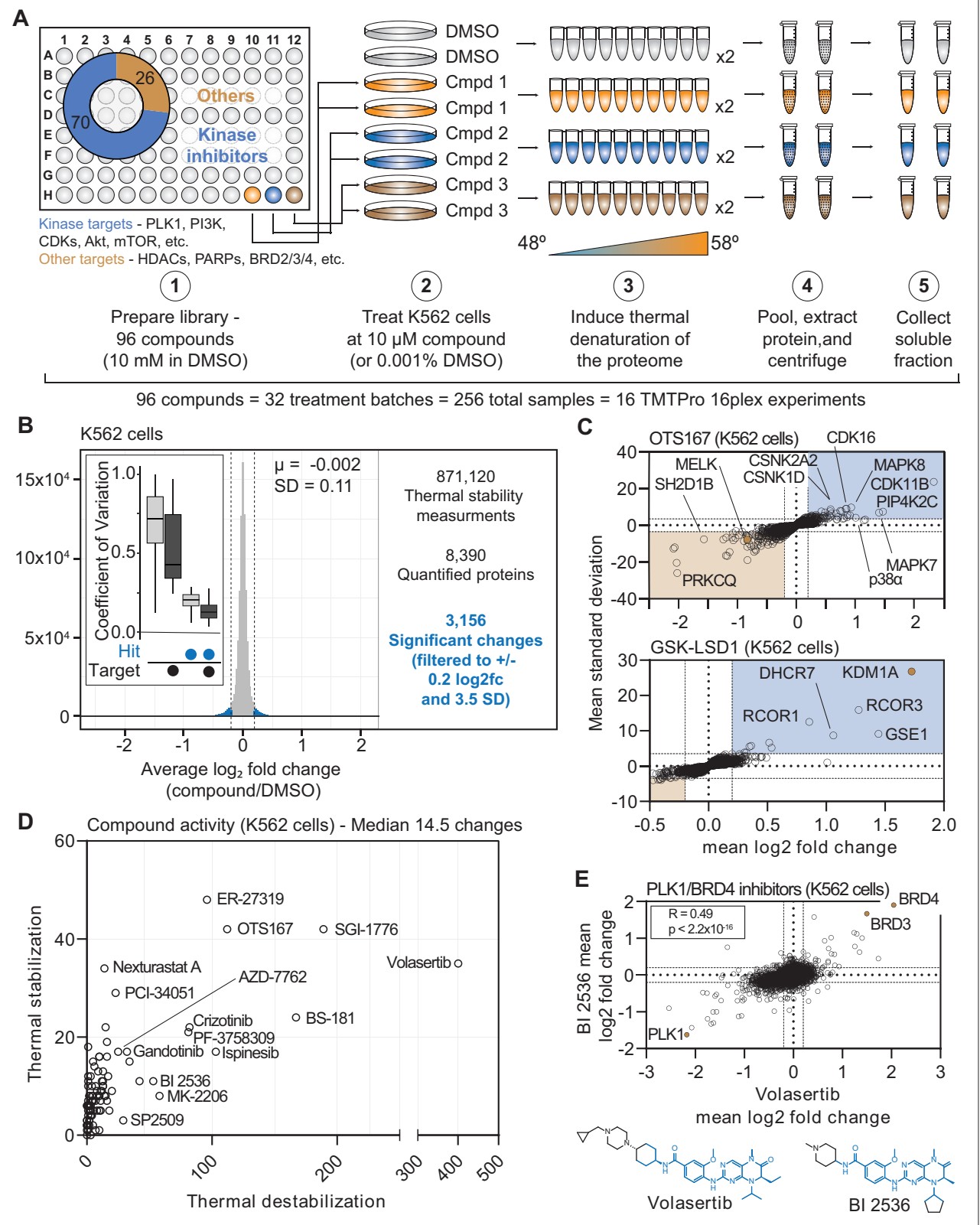

**Figure 1.** Establishing a robust and efficient workflow for assessing changes in protein thermal stability in living cells. (**A**) Schematic of the cell-based PISA workflow utilized in the large-scale chemical perturbation screen. (**B**) Histogram plotting the log$_2$ fold change values for all solubility (thermal stability) measurements (main panel). Box plot depicting the %CV of curated targets vs. all other proteins (inset). (**C**) Plots highlighting the significant changes in protein solubility following treatment with OTS167 (top) or GSK-LSD1 (bottom). The mean log$_2$ fold change of duplicate measurements is

*Figure 1 continued on next page*

*Figure 1 continued*

plotted on the x-axis and the mean nSD is plotted on the y-axis. Blue boxes contain proteins that exhibit an increase in solubility (increase in melting temperature) and orange boxes contain proteins that exhibit a decrease in solubility (decrease in melting temperature). Orange points represent the known target of each compound. (**D**) The total number of significant solubility changes quantified upon treatment with each compound. The number of stabilizing events (increase in solubility) is plotted on the y-axis and destabilizing events (decrease in solubility) are plotted on the x-axis. (**E**) Plot comparing the log₂ fold changes measurements for each protein in BI 2536 (y-axis)- and volasertib (x-axis)-treated K562 cells.

The online version of this article includes the following source data and figure supplement(s) for figure 1:

**Source data 1.** This table contains cell-based PISA data for K562 cells treated with ribociclib, palbociclib, and abemaciclib.

**Source data 2.** This three part table contains the PISA data for cell- and lysate-based screens (part 1 of 3 for this table).

**Source data 3.** This three part table contains the PISA data for cell- and lysate-based screens (part 2 of 3 for this table).

**Source data 4.** This three part table contains the PISA data for cell- and lysate-based screens (part 3 of 3 for this table).

**Figure supplement 1.** Establishing a robust and efficient workflow for assessing changes in protein thermal stability in living cells.

**Figure supplement 2.** Establishing a robust and efficient workflow for assessing changes in protein thermal stability in living cells.

**Figure supplement 3.** Establishing a robust and efficient workflow for assessing changes in protein thermal stability in living cells.

**Figure supplement 4.** Establishing a robust and efficient workflow for assessing changes in protein thermal stability in living cells.

**Figure supplement 5.** Establishing a robust and efficient workflow for assessing changes in protein thermal stability in living cells.

which were arranged into 16 TMTpro 16-plex experiments and required a minimum of 384 hr of instrument time (*Figure 1—figure supplement 3A*).

In total, we quantified approximately 6,800 proteins per treatment for a total of 871,120 drug-protein thermal stability measurements (*Figure 1B*, *Figure 1—figure supplement 3B*, *Supplementary file 3*, and *Figure 1—source data 2*). Compared to the full dataset, annotated targets of the library compounds had significantly different solubility (thermal stability) compared to non-targets (Wilcoxon rank sum test, $p=3.56 \times 10^{-17}$). To define high-confidence thermal stability changes, we developed an empirically derived framework (see Methods; *Figure 1—figure supplement 3C*). Briefly, we determined the log₂ fold change of each compound treatment in reference to the vehicle-treated control and then quantified the trimmed standard deviation of all solubility measurements for a given protein across all treatments. In order to be considered a hit, the log₂ fold change needed to be 3.5 standard deviations from the mean and have an absolute value greater than or equal to 0.2 ($|\log_{2Cmpd/DMSO}|>0.2$, $|nSD|>3.5$; *Figure 1—figure supplement 3C–G*). The fold change filtering removed background proteins while the per protein standard deviation filtering removed proteins that were observed to engage in large numbers of non-specific interactions, such as the kinase GAK (*Figure 1—source data 2*). While, we believe this strategy is capable of identifying meaningful compound-dependent changes, there are examples of compound-target pairs that surpass the standard deviation cutoff, but not the log₂ fold change cutoff and vice versa. This suggests that this filtering strategy might be overly conservative (*Figure 1—figure supplement 3F–H*). Nonetheless, from the 871,120 total protein solubility measurements, filtering the data resulted in 3156 putative hits ($|\log_{2Cmpd/DMSO}|>0.2$, $|nSD|>3.5$; *Figure 1B*), each of which point to a compound-dependent change in protein thermal stability.

Upon assembling compound-centric PISA solubility (thermal stability) profiles, known compound targets could be easily resolved. For example, in cells treated with OTS167, the primary target, MELK, had one of the largest negative log₂ fold changes ($\log_{2Cmpd/DMSO}$: –0.840; nSD: –5.688), which is consistent with a thermal destabilization (*Figure 1C*). In addition to its intended target, OTS167 also induced consistent protein solubility changes in other kinases (WEE1, PRKD3, CDK11B, AURKA, and others). These findings are consistent with previous reports demonstrating that the mechanism of action for OTS167 is in part driven by off-target protein engagement (*Giuliano et al., 2018*). Similarly, treatment with GSK-LSD1 caused an increase in the solubility of its primary target—the lysine demethylase, KDM1A, which likely results from a compound-dependent thermal stabilization ($\log_{2Cmpd/DMSO}$: 1.726; nSD: 26.825; *Figure 1C*). In addition to the primary target, GSK-LSD1 also impacted several proteins known to form a complex with KDM1A, including GSE1 and RCOR1/3 (*Figure 1C*). Finally, we observed disparities in how different molecules targeting the same protein impacted the target. The library contained two compounds known to target AKT1—MK-2206, an allosteric inhibitor and CCT128930, an ATP-competitive inhibitor (*Amiran et al., 2023*; *Hirai et al., 2010*; *Yap et al., 2011*). Interestingly, engagement of the allosteric ligand, MK-2206, induced a large change in AKT1

solubility, while the ATP-competitive inhibitor appeared to have no effect (*Figure 1—figure supplement 4A and B*).

Beyond the known targets, these data define a set of high-confidence ligand-induced thermal stability changes for each of the 96 compounds assayed in this screen (*Figure 1D*). OTS167 caused a significant change in the solubility (thermal stability) of 154 proteins, including other protein kinases, which likely stem from off-target engagement (*Figure 1C and D*). GSK-LSD1 treatment, on the other hand, resulted in only ten filtered changes (*Figure 1C and D*). Although OTS167 was among the most active compounds in the screen (as defined by total filtered changes), treatment with the PLK1 inhibitor volasertib resulted in the most changes (435, *Figure 1D*). While there were several other compounds that caused >100 changes in (ER-27319, BS-181, SGI-1776, etc.), the median number of filtered changes per compound was 14.5 (*Figure 1D*).

Our screening library contained two dual PLK1/BRD4 inhibitors—volasertib and BI 2536—which have similar chemical structures (*Figure 1E*). Appropriately, the PISA solubility profiles that resulted from treatment with each compound correlated well ($r_{spearman}$ = 0.49, p<1.0 × 10$^{-15}$), highlighted by consistent changes in the solubility (thermal stability) of PLK1, BRD4, and the proteome in general (*Figure 1E*). In addition to these dual PLK1/BRD4 inhibitors, the library also contained two compounds—JQ-1 and CPI-203—which target BRD4, but not PLK1. While the PISA profiles of the dual PLK1/BRD4 inhibitors were correlated, this was not the case for volasertib and JQ-1. Indeed, there was clear evidence that JQ-1 could engage BRD4 but not PLK1 however, there was little overall correlation, suggesting that the observed volasertib-dependent changes are largely PLK1-dependent (*Figure 1—figure supplement 4C*).

Finally, we wanted to determine if the set of high-confidence solubility (thermal stability) alterations could provide insights related to compound mechanism of action. PF-3758309 is designated as a PAK4 inhibitor. Including PAK4, treatment with this inhibitor caused a total of 102 proteins to undergo a significant change in solubility (*Figure 1D*). Expectedly, this list included a number of protein kinases including MELK, TBK1, WEE1, and others (*Figure 1—figure supplement 5A*). Unexpectedly, there was also a significant enrichment of spliceosome subunits among these proteins (*Figure 1—figure supplement 5A*). These changes could stem from upstream inhibition of a kinase that regulates splicing and that the mechanism of action of this compound might, at least, in part, works through an alteration of this process. This result is consistent with a recent study that identified PF-3758309 as a potent modulator of pre-mRNA splicing (*Shi et al., 2020*). Overall, these data provide further evidence that PF-3758309 might be capable of impacting the assembly and/or activity of the spliceosome.

## A protein-centric view of compound-dependent changes in thermal stability

Having established criteria for identifying significant compound-dependent changes in protein thermal stability, we focused on the known targets of the library compounds in order to identify key factors affecting the interpretation of our screening data. In total, we quantified at least one known target for 79 of the 96 compounds assayed in the primary screen (84%) and found that 56 (71%) of these compounds induced a change in the solubility (thermal stability) of an annotated target (*Figure 2A*). While compound treatment more commonly resulted in an increase in the solubility (thermal stabilization) of known targets, we also observed a number of examples in which inhibitor binding stimulated a decrease in solubility (thermal destabilization). A consistent increase in solubility was observed for proteins including BRD4, HDAC1, and AURKA. Conversely, compound engagement by PLK1 resulted in a consistent decrease in solubility (*Figure 2A*). Therefore, both positive and negative log$_2$ fold changes represent evidence of target engagement and specific proteins are consistently impacted by inhibitor binding. This result is consistent with previous studies and serves to corroborate the quality of our screening data (*Sabatier et al., 2022*).

Next, we characterized the magnitude of solubility (thermal stability) changes that we measured for known targets. Having treated cells with each compound at 10 μM (a concentration commonly used in similar studies; *Savitski et al., 2014*; *Gaetani et al., 2019*; *Mitchell et al., 2023*), we assumed that each target was fully saturated with a given compound. Therefore, the measurements likely reflect the maximal ligand-induced solubility (thermal stability) changes for each target. We observed a large range of solubility measurements for known compound-target pairs, from a four-fold reduction in protein solubility after thermal denaturation to a four-fold increase in protein solubility upon

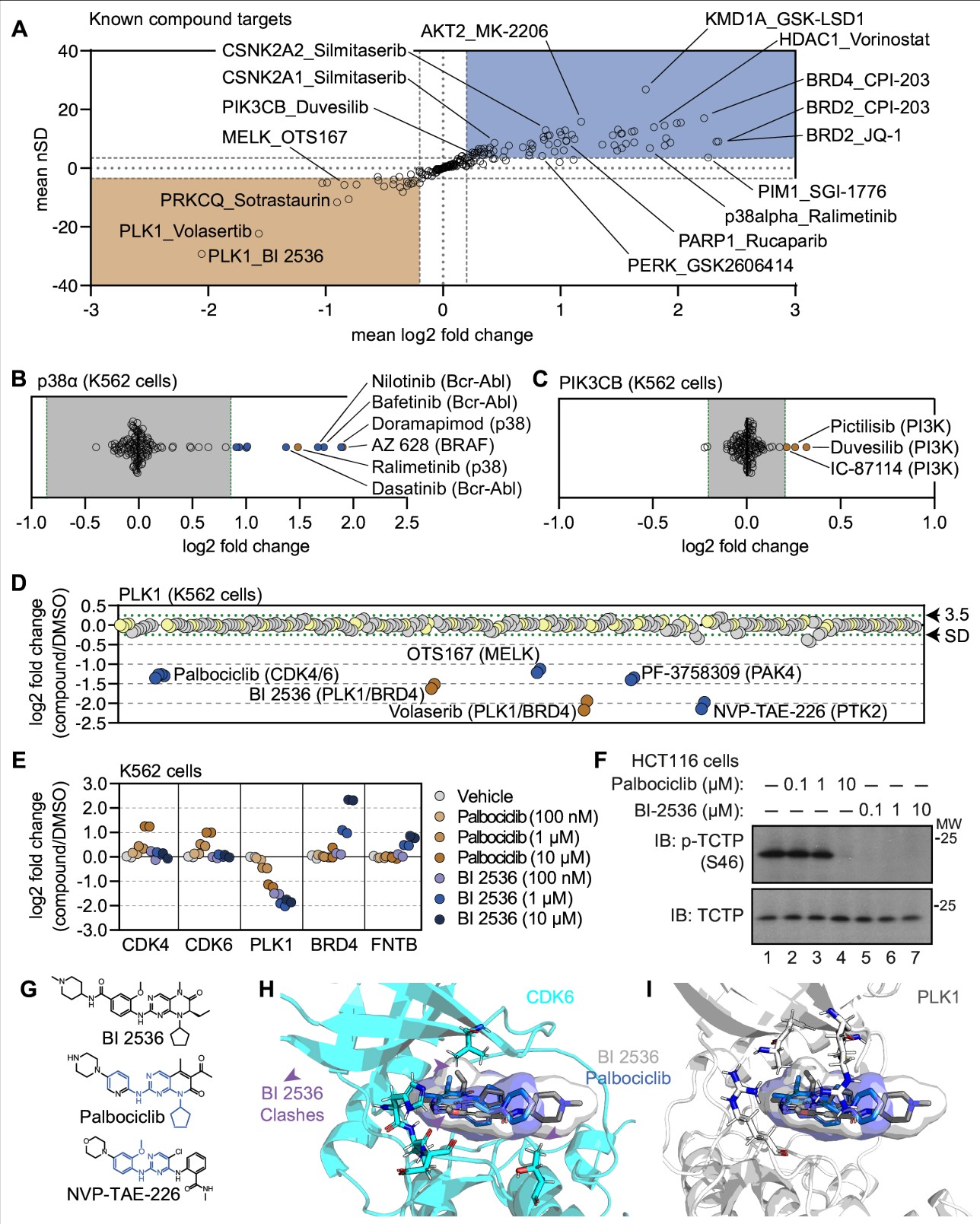

**Figure 2.** Using cell-based PISA data to assess compound target engagement. (**A**) A plot of the mean $\log_2$ fold change (x-axis) and mean nSD (y-axis) of each known compound-target pair that were quantified in the screen. Blue boxes contain proteins that exhibit an increase in solubility (increase in melting temperature) and orange boxes contain proteins that exhibit a decrease in solubility (decrease in melting temperature). (**B** and **C**) Plots depicting the mean $\log_2$ fold change of p38α (**B**) and PIK3CB (**C**) in response to each of the compound and DMSO treatments. Orange points indicate

*Figure 2 continued on next page*

*Figure 2 continued*

compounds known to target each protein. Blue points indicate other compounds that induce a significant change in thermal stability for each protein. Green dashed lines mark a SD of 3.5 for each protein. (**D**) A protein-centric view of PLK1 solubility (thermal stability) in response to all treatments. $Log_2$ fold change is plotted on the y-axis. The points represent each of the 256 treatments that were performed. Orange points indicate compounds known to target PLK1. Blue points indicate other compounds that result in a significant decrease in solubility of PLK1. Green dashed lines mark a SD of 3.5 cutoff for PLK1. (**E** and **F**) K562 cells were treated with the indicated concentrations of palbociclib or BI 2536 for 15 minutes. Changes in protein thermal stability are represented as a $log_2$ fold change in soluble protein abundance for each treatment in reference to a DMSO-treated control. (**E**) PLK1 activity was determined using a western blot for p-TCTP (S46) and total TCTP (**F**). (**G**) Chemical structures of BI 2536, palbociclib and NVP-TAE-226. Common structural features with BI 2536 are highlighted in blue for Palbociclib and NVP-TAE-226. (**H**) Co-crystal structure of CDK6 bound to palbociclib (blue) with BI 2536 (gray) modeled into the active site. (**I**) Co-crystal structure of PLK1 bound to BI 2536 (gray) with palbociclib (blue) modeled into the active site.

The online version of this article includes the following source data and figure supplement(s) for figure 2:

**Source data 1.** This table contains cell-based PISA data for K562 cells treated with increasing concentrations of BI 2536 and palbocilicb.

**Source data 2.** This table contains cell-based PISA data for K562 cells treated with increasing concentrations of BI 2536 and NVP-TAE-226.

**Source data 3.** PDF file containing original western blots for ***Figure 2F***, indicating the relevant bands and treaments.

**Source data 4.** Original files for western blot analysis displayed in ***Figure 2F***.

**Source data 5.** PDF file containing original western blots for ***Figure 2—figure supplement 1D***, indicating the relevant bands and treatments.

**Source data 6.** Original files for western blot analysis displayed in ***Figure 2—figure supplement 1D***.

**Figure supplement 1.** Using cell-based PISA data to assess compound target engagement.

compound engagement (***Figure 2A***). We also observed many proteins with small (15%) but consistent solubility changes following compound treatment. p38α-targeting compounds ralimetinib and doramapimod caused a large apparent increase in p38α solubility (thermal stability; $log_{2Cmpd/DMSO}$ = 1.52 and 1.94, respectively), while compounds targeting PI3K (pictilisib, duvesilib, and IC-87114), on the other hand, caused smaller solubility (thermal stability) changes ($log_{2Cmpd/DMSO}$ = 0.258, 0.319, and 0.213, respectively; ***Figure 2B and C*** and ***Figure 2—figure supplement 1A and B***). In all cases, each of these measurements surpassed the nSD cutoff of 3.5 for each particular target. Taken at face value, one might conclude that the greater log2 fold change in solubility that doramapimod exerts on p38α is more meaningful than the much smaller change that pictilisib exerts on PIK3CB, however, this is not necessarily true as both compounds can engage their targets at nanomolar concentrations. Instead, our data appears to be consistent with the previous observation that the maximum ligand-induced change in thermal stability is target-specific (***Savitski et al., 2014***; ***Becher et al., 2016***). Therefore, a small log2 fold change for one protein (PIK3CB, $log_{2Cmpd/DMSO} \geq 0.2$), can be just as meaningful as a large log2 fold change for another (p38α, $log_{2Cmpd/DMSO} \geq 2$).

## Using the cell-based screening data to identify off-target compound engagement

Having defined a set of high-confidence solubility (thermal stability) changes for each compound, we next looked for potential examples of off-target engagement. In addition to volasertib and BI 2536 (PLK1/BRD inhibitors), several other compounds from the screen also impacted the solubility of PLK1 to a similar extent (***Figure 2D***). These included two promiscuous kinase inhibitors—OTS167 and PF-3758309. In addition to these compounds, two other ATP-competitive kinase inhibitors also impacted the solubility of PLK1—NVP-TAE-226, a FAK inhibitor and palbociclib, one of the CDK4/6 inhibitors used to establish our workflow. Because these two compounds induced a change in solubility to a similar magnitude as BI 2536 and volasertib, we hypothesized that these molecules might bind the active site and inhibit PLK1.

To further explore the impact of NVP-TAE-226 and palbociclib on PLK1 thermal stability, we treated K562 cells with increasing concentrations of BI 2536, NVP-TAE-226, and palbociclib, ranging from 100 nM to 10 μM and performed cell-based PISA profiling to approximate the changes in thermal stability that occurred at each inhibitor concentration (***Figure 2—source data 1 and 2***). As supported by previous reports, this would provide a relative assessment of the ability of each molecule to engage (and inhibit) PLK1 (***Savitski et al., 2014***; ***Becher et al., 2016***; ***Mateus et al., 2016***; ***Sabatier et al., 2022***). Appropriately, BI 2536 caused a dose-dependent decrease in PLK1 solubility down to the lowest dose tested (100 nM; ***Figure 2E*** and ***Figure 2—figure supplement 1C***). Treatment with

palbociclib and NVP-TAE-226 also caused a dose-dependent decrease in PLK1 solubility, however, each of these compounds appeared to impact PLK1 to a lesser extent than BI 2536. In fact, 100 nM BI 2536 treatment impacted PLK1 to the same extent as 10 μM treatments with palbociclib and NVP-TAE-226 (*Figure 2E* and *Figure 2—figure supplement 1C*). Next, we synchronized HCT116 cells in G2/M phase to test the ability of each compound to directly inhibit PLK1 in a cell-based assay. During this phase of the cell cycle, PLK1 is active and known to phosphorylate TCTP (at serine 46), a protein involved in microtubule stabilization (*Cucchi et al., 2010*). Following synchronization, these cells were treated with increasing concentrations of each compound and phosphorylation of TCTP at S46 was assayed to read out PLK1 activity. BI 2536 completely abolished PLK1-dependent phosphorylation of TCTP at all tested doses. Palbociclib and NVP-TAE-226 also inhibited TCTP phosphorylation, however, these compounds were much less potent than BI 2536 and were only able to inhibit PLK1 at doses between 1 μM and 10 μM (*Figure 2F* and *Figure 2—figure supplement 1D*). Importantly, these results closely mirror the PISA data, providing further evidence that proteomic thermal stability measurements accurately recapitulate target engagement. Therefore, palbociclib and NVP-TAE-226 appear to be weak inhibitors of PLK1, which is consistent with previous studies that suggested these compounds as off-target inhibitors of this kinase (*Hafner et al., 2019*).

The PISA data suggests that palbociclib was able to engage both CDK4/6 and PLK1, while BI 2536 was only able to engage PLK1 (*Figure 2E*). This is despite the fact that both compounds have a similar chemical structure (*Figure 2G*). In order to further interrogate this disparity, we modeled palbociclib and BI 2536 into co-crystal structures of PLK1 (PDB: 2rku) and CDK6 (PDB: 5l2i) with their 'specific' inhibitors (*Figure 2H1*; *Chen et al., 2016*; *Kothe et al., 2007*). When we overlaid BI 2536 in the active site of CDK6, we observed three points of steric clashing with the protein density of the CDK6 active site. These findings were consistent with the lack of CDK4/6 solubility changes in cells that were treated with BI 2536. From overlaid projections of palbociclib and BI 2536 in PLK1, we did not observe steric interference in binding suggesting that both of these inhibitors could readily bind to the active site of PLK1. These data are again consistent with our observed decrease in solubility of PLK1 in cells that were treated with either BI 2536 or palbociclib.

## PISA screening in crude cell extracts

Thus far we have focused exclusively on living cells, but PISA can also be used to screen for changes in cell lysates or crude extracts (*Molina et al., 2013*; *Savitski et al., 2014*; *Franken et al., 2015*; *Becher et al., 2016*; *Sridharan et al., 2019b*; *Liang et al., 2022*). Having already assembled one of the largest drug-based thermal denaturation studies in cells, we wanted to further profile a subset of these compounds in native extracts. We selected 70 compounds from the cell-based screen for further interrogation in native cell lysates. For this screen, we prepared crude extracts by dounce homogenizing K562 cells in PBS (*Sridharan et al., 2019b*). The crude extracts were treated with each of the 70 compounds in biological duplicate at 10 μM for 30 min and any changes in thermal stability were approximated using PISA. In total, the 70 compounds and DMSO controls generated a total of 160 samples, which were arrayed across 10 TMT 16-plex experiments and quantified using approximately 240 hr of instrument time (*Figure 3A* and *Figure 3—figure supplement 1A*).

We quantified an average of 7840 proteins per treatment (*Figure 3B* and *Figure 3—figure supplement 1B*), which resulted in a total of 627,176 solubility (thermal stability) measurements from all K562 lysate-based experiments (*Figure 3B*, *Supplementary file 4*, and *Figure 1—source data 2*). After applying the same filters used in the cell-based screen, our dataset consisted of 2176 protein solubility (thermal stability) changes (*Figure 3B* and *Figure 3—figure supplement 1C*). Similar to the cell-based studies, the known targets of library compounds frequently experienced a significant change in thermal stability (*Figure 3—figure supplement 1C–G*). For example, treatment of K562 lysates with MK-2206 and GSK-LSD1 increased the solubility (thermally stabilized) AKT1/2 and KDM1A, respectively (*Figure 3C*). In addition to AKT1/2, MK-2206 also increased the solubility of additional kinases, including subunits of RPS6K and PI3K, which highlights the ability of this approach to find putative examples of off-target engagement (*Figure 3C*). Finally, we observed a GSK-LSD1-dependent changes in the solubility of several KDM1 binding partners, which echoes the similar observation made in cell-based experiments (*Figure 3C*).

Since changes in thermal stability in lysate-based experiments stem almost exclusively from direct ligand binding events, we expected to observe fewer overall solubility (thermal stability) changes in

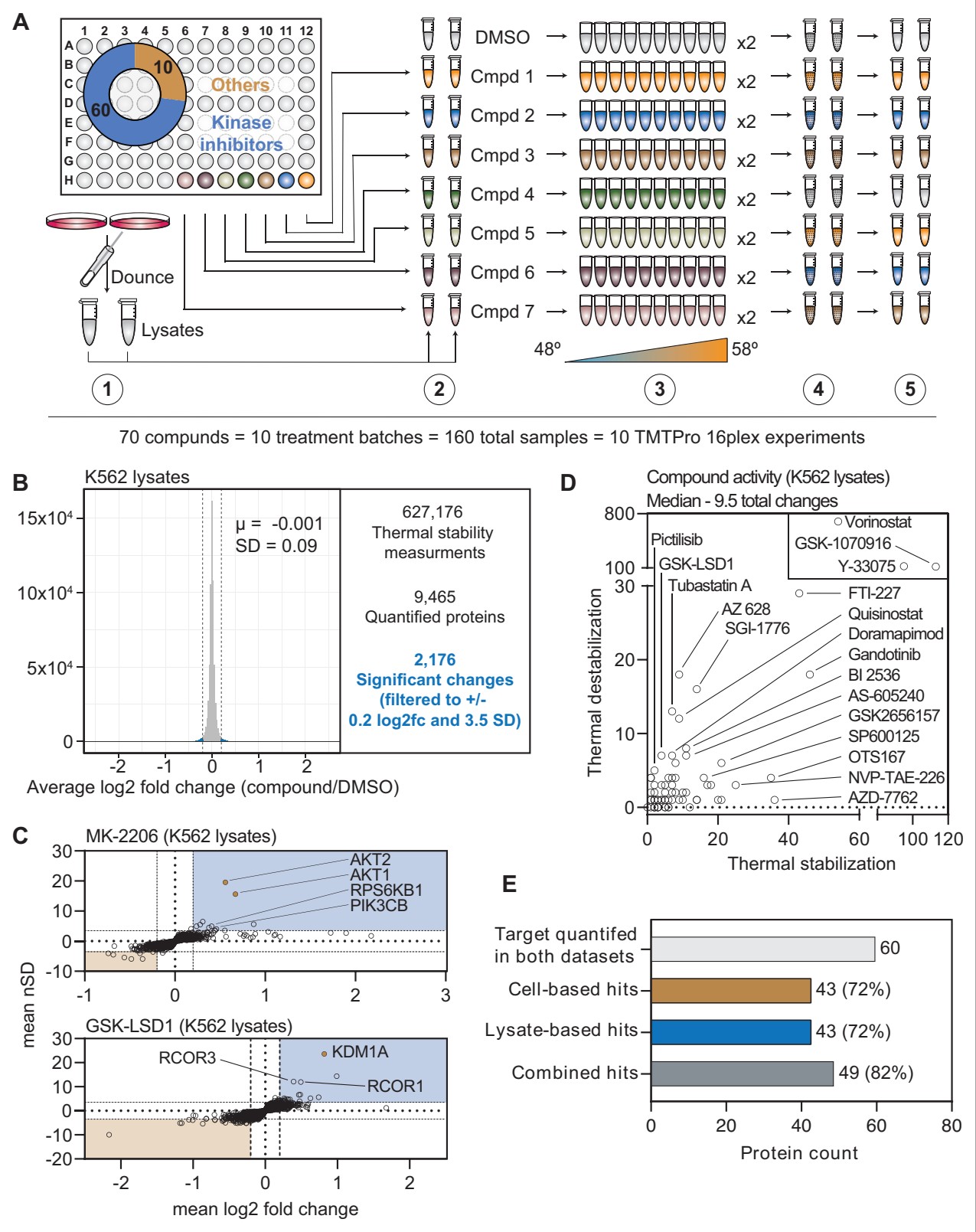

**Figure 3.** A chemical perturbation screen of 70 compounds in K562 native extracts. (**A**) Schematic of the lysate-based PISA workflow utilized in the large-scale chemical perturbation screen. (**B**) Histogram plotting the $log_2$ fold change values for all solubility measurements in lysate-based screen. (**C**) Plots highlighting the significant changes in protein solubility (thermal stability) following treatment with MK-2206 (top) or GSK-LSD1 (bottom). The mean $log_2$ fold change of duplicate measurements is plotted on the x-axis and the mean nSD is plotted on the y-axis. Blue boxes contain proteins that exhibit

*Figure 3 continued*

an increase in solubility (increase in melting temperature) and orange boxes contain proteins that exhibit a decrease in solubility (decrease in melting temperature). Orange points represent the known target of each compound. (**D**) Plot depicting the total solubility changes quantified upon treatment with each compound. The number of stabilizing events (increase in solubility) is plotted on the x-axis and destabilizing events (decrease in solubility) are plotted on the y-axis. (**E**) Plot depicting the number of compounds for which a target was quantified in both cell- and lysate-based experiments (light gray), the number of compounds that caused a significant change in a known target in cell-based experiments (orange), the number of compounds that caused a significant change in a known target in lysate-based experiments (blue), and the number of compounds that caused a significant change in a known target in cell- or lysate-based experiments.

The online version of this article includes the following figure supplement(s) for figure 3:

**Figure supplement 1.** A chemical perturbation screen of 70 compounds in K562 native extracts.

**Figure supplement 2.** A chemical perturbation screen of 70 compounds in K562 native extracts.

this setting compared to cell-based experiments (*Savitski et al., 2014*; *Franken et al., 2015*; *Huber et al., 2015*; *Becher et al., 2016*). As expected, the lysate-based screen (2176) yielded fewer significant changes than the cell-based screen (3156). It is also important to note that three compounds (vorinostat, Y-33075, and GSK-1070916) contribute 1200 (55%) of all the significant changes in the lysate-based screen. It is highly unlikely that these three molecules actively engage so many proteins and, therefore, the 2176 hits in the lysate-based screen were likely affected in part by consistent, but artefactual effects of lysate-based analyses that we do not fully understand (*Van Vranken et al., 2021*). Overall, the median number of protein solubility changes in lysates was 66% of those observed in cells, with 9.5 proteins in lysate-based experiments compared to 14.5 in cell-based experiments (*Figure 3D*).

In order to directly compare the ability of each approach to identify an on-target hit, we focused on the 60 compounds for which a known target was quantified in both cell- and lysate-based experiments. Of these 60 compounds, both approaches identified a change in a known target of 43 compounds (~72%; *Figure 3E*). Importantly, these were not the same 43 compounds (*Figure 3—figure supplement 2A*). For example, an alisertib-dependent change in AURKA solubility occurred in cells, but not lysates, while an AZ 628-dependent change in BRAF solubility was observed in lysates, but not cells (*Figure 3—figure supplement 2A and B*). Consideration of each dataset in isolation results in an on-target hit rate of ~72%. Combing the datasets, on the other hand, results in an on-target hit rate of ~82% (49/60; *Figure 3E* and *Figure 3—figure supplement 2A*). Therefore, combining the two approaches increases the chances of finding the target of a compound of interest.

## Cell- and lysate-based PISA are complimentary approaches to determine compound engagement

Having re-screened dozens of compounds in lysate-based experiments, we wanted to assess the complementarity of these approaches. We focused on the known compound targets quantified in both cell- and lysate-based experiments and compared the solubility (thermal stability) changes measured using each approach (*Figure 4A*). Encouragingly, we observed compounds that impacted their known targets in both cell- and lysate-based approaches. These include GSK-1070916 and MK-2206, which increased the solubility of AURKA and AKT1, respectively, in both cell- and lysate-based experiments (*Figure 4A and B*). Conversely, we observed compounds that failed to impact the solubility of their target proteins in either the cell-based or lysate-based experiments (*Figure 4A* and *Figure 4—figure supplement 1A*). These include compounds such as IRAK4-Inhibitor-1 and entospletinib which are known to target IRAK4 and SYK kinase, respectively. We note that the magnitudes of the solubility changes varied between the cell- and lysate-based approaches. Yet, compounds that increased the solubility of their annotated protein targets in cells also tended to increase the solubility their targets in lysates with one notable exception. Sotrastaurin induces an increase in the solubility (thermal stabilization) of its target, PRKCA, in lysates ($\log_{2Cmpd/DMSO}$ = 0.924), but a decrease in the solubility (thermal destabilization) in cells ($\log_{2Cmpd/DMSO}$ = -0.502; *Figure 4A and B*). These data suggest that the decrease in solubility observed in cell-based experiments might stem from a complex biophysical rearrangement.

While many compounds significantly altered their known targets using both approaches, there were some compounds that preferentially altered the solubility (thermal stability) of known targets in one experimental setting but not the other (*Figure 4A*). For example, there were four compounds known

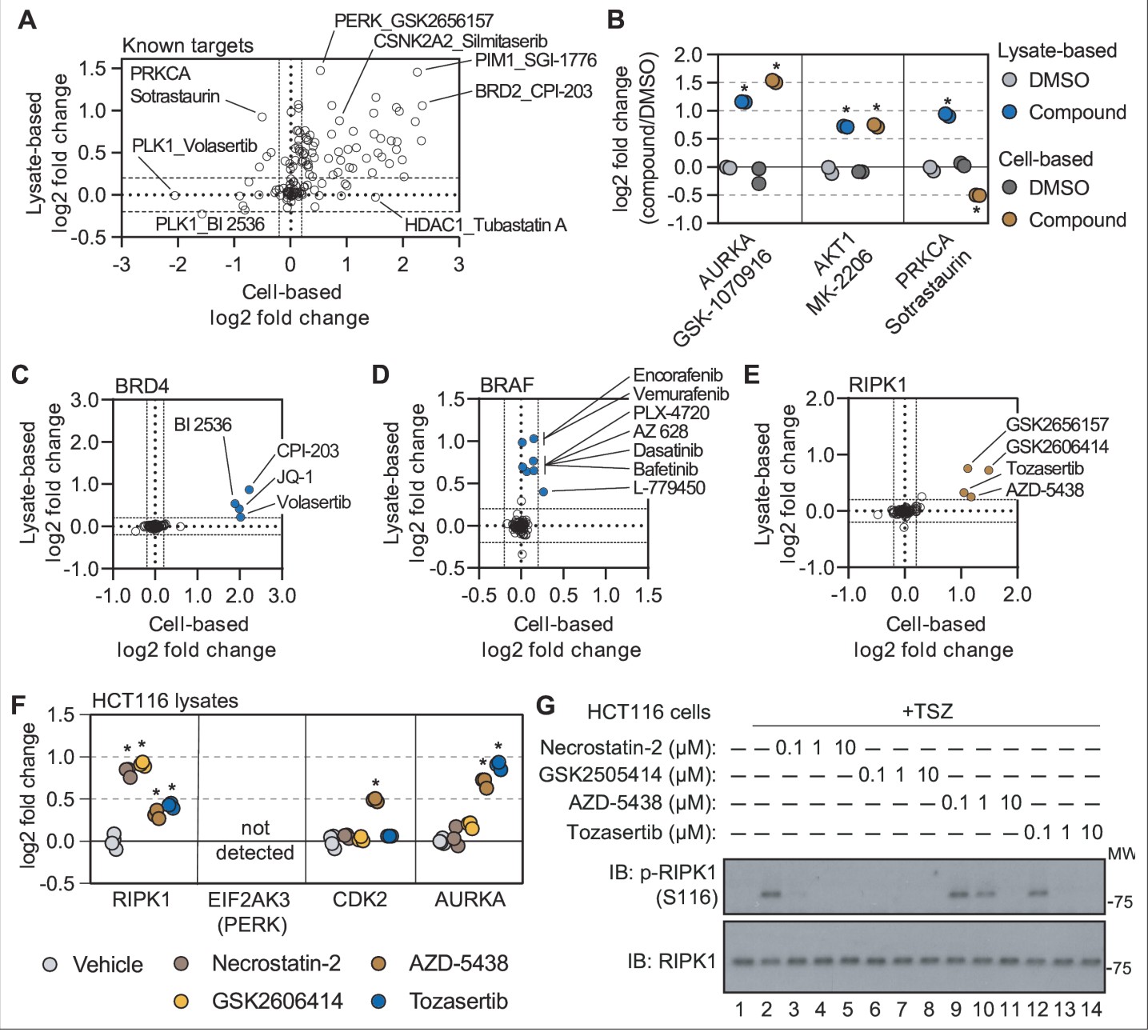

**Figure 4.** Cell- and lysate-based PISA are complimentary approaches for assessing mechanism of action. (A) A plot depicting the mean log₂ fold change for all compound-target pairs quantified in cell- (x-axis) and lysate-based (y-axis) PISA. Dashed lines indicate a log₂ fold change of +/-0.2. (B) The log₂ fold change of a selected number of compound-target pair in cell- and lysate-based PISA. (C-E) Plots depicting the log₂ fold change values for BRD4 (C), BRAF (D), and RIPK1 (E) in response to treatment with all compounds assayed using cell- and lysate-based PISA. Dashed lines indicate a log₂ fold change of +/-0.2. (F) HCT116 cell lysates (N=3) were treated with each compound or DMSO for 15 min and assay using PISA. The plot indicates the log₂ fold change of several proteins in response to treatment with each compound. (G) HCT116 cells were treated with the indicated concentration of each compound for 1 hr following the initiation of necroptosis using TSZ treatment. RIPK1 autophosphorylation was assayed by western blot using antibodies targeting p-RIPK1 (S116) or total RIPK1.

The online version of this article includes the following source data and figure supplement(s) for figure 4:

**Source data 1.** This table contains lysate-based PISA data for HCT116 lysates treated with necrostatin-2, GSK2606414, AZD-5438, and tozasertib.

**Source data 2.** PDF file containing original western blots for *Figure 4G*, indicating the relevant bands and treatments.

**Source data 3.** Original files for western blot analysis displayed in *Figure 4G*.

**Figure supplement 1.** Cell- and lysate-based PISA are complimentary approaches for assessing mechanism of action.

to target BRD4 (JQ-1, CPI-203, BI 2536, and volasertib) that were assayed in both cell- and lysate-based experiments. Treatment of cells with any of these four inhibitors resulted in a $\log_{2Cmpd/DMSO} \sim 2$, while the lysate-based assays resulted in smaller changes (*Figure 4C* and *Figure 4—figure supplement 1A*). Conversely, multiple inhibitors caused a significant change in BRAF solubility in lysates (AZ 628, PLX-4720, L-779450) but did not impact BRAF in cell-based assays (*Figure 4D* and *Figure 4—figure supplement 1A*). Thus, certain protein targets were more prone to solubility (thermal stability) changes in one experimental setting compared to the other (*Huber et al., 2015*).

## Combining cell- and lysate-based data to discover off-target engagement

Having demonstrated the complementarity of the cell- and lysate-based approaches, we sought to explore the corroborative value of integrating these data. We reasoned that any unexpected ligand-induced changes in solubility (thermal stability) that were shared between the two approaches would provide strong evidence of compound engagement. Despite an absence in the library of compounds designated as RIPK1 kinase inhibitors, there were four compounds that significantly increased the solubility of RIPK1 in cells and lysates (*Figure 4E* and *Figure 4—figure supplement 1B*). Although three of the compounds—GSK-2656167, GSK-2606414, and tozasertib—had previously been shown to be off-target inhibitors of RIPK1, AZD-5438, a cyclin-dependent kinase (CDK) inhibitor, had never been attributed such activity (*Byth et al., 2009*; *Martens et al., 2018*; *Rojas-Rivera et al., 2017*). Nonetheless, this compound was capable of increasing RIPK1 solubility to a similar extent as the other three known RIPK1 off-target inhibitors (*Figure 4E* and *Figure 4—figure supplement 1B*).

In order to gain a sense of the potency of AZD-5438 (and the other molecules), we assayed these compounds alongside necrostatin-2, a bona fide RIPK1 inhibitor, in HCT116 lysates (*Figure 4—source data 1*). Treatment with necrostatin-2 or GSK-2606414 resulted in a similar solubility change for RIPK1, that was far greater than both tozasertib and AZD-5438 (*Figure 4F* and *Figure 4—figure supplement 1C–F*). Treatment of HCT116 cells with TNFα, a Smad mimetic, and a pan-caspase inhibitor (zVAD) will lead to an initiation of necroptosis and, importantly, autophosphorylation of S116 by RIPK1 (*Laurien et al., 2020*). While each of our putative RIPK1 inhibitors was capable of inhibiting this autophosphorylation, AZD-5438 and tozasertib were less potent than necrostatin-2 and GSK-2606414 (*Figure 4G*). Therefore, the thermal stability measurements made in our lysate-based PISA experiment correlated with compound potency. This demonstrates that the combination of cell- and lysate-based screening data can be employed to identify instances of off-target engagement, even for weak inhibitors.

In addition to compound potency, the combined cell and lysate data provided valuable information regarding compound specificity. While it is difficult to separate necrostatin-2 and GSK-2606414 based on potency, necrostatin-2 had a greater apparent specificity with respect to RIPK1 engagement. Indeed, necrostatin-2 only impacted the solubility of RIPK1 in the lysate-based experiments (*Figure 4—figure supplement 1C*). GSK-2606414, on the other hand, engaged other kinases including CSK, LIMK1, and TBK1 (*Figure 4—figure supplement 1D*). Finally, AZD-5438 and tozasertib were less potent and less specific for RIPK1 than necrostatin-2 (*Figure 4—figure supplement 1E and F*).

## Disparities in cell- and lysate-based data pinpoint secondary changes in protein thermal stability

In comparison to lysate-based approaches, cell-based experiments have added potential to identify secondary changes in protein thermal stability that occur independent of direct ligand binding (*Savitski et al., 2014*; *Almqvist et al., 2016*; *Becher et al., 2016*; *Sridharan et al., 2019b*; *Liang et al., 2022*; *Becher et al., 2018*; *Dai et al., 2018*). These changes could occur as a result of changes in interactions with other proteins, ligands, nucleic acids, or the effects of PTMs. Importantly, all these factors can help to define compound mechanism of action. This type of change was previously highlighted in the context of the CDK4/6 inhibitors. Indeed, treatment of cells with CDK4/6 inhibitors induced a decrease in the solubility (thermal destabilization) of RB1, which we attributed to a change in phosphorylation due to upstream target inhibition (*Figure 1—figure supplement 1C*). Other CDK inhibitors (AZD-5438 and flavopiridol) also had a similar effect on RB1. In lysates, on the other hand, when cellular signaling networks are disrupted, RB1 was unaffected by these compounds (*Figure 5A*). This concept is also apparent when comparing the PF-3758309-dependent PISA profiles in each experimental setting (*Figure 5—figure supplement 1A*). Indeed, there is a striking difference in the

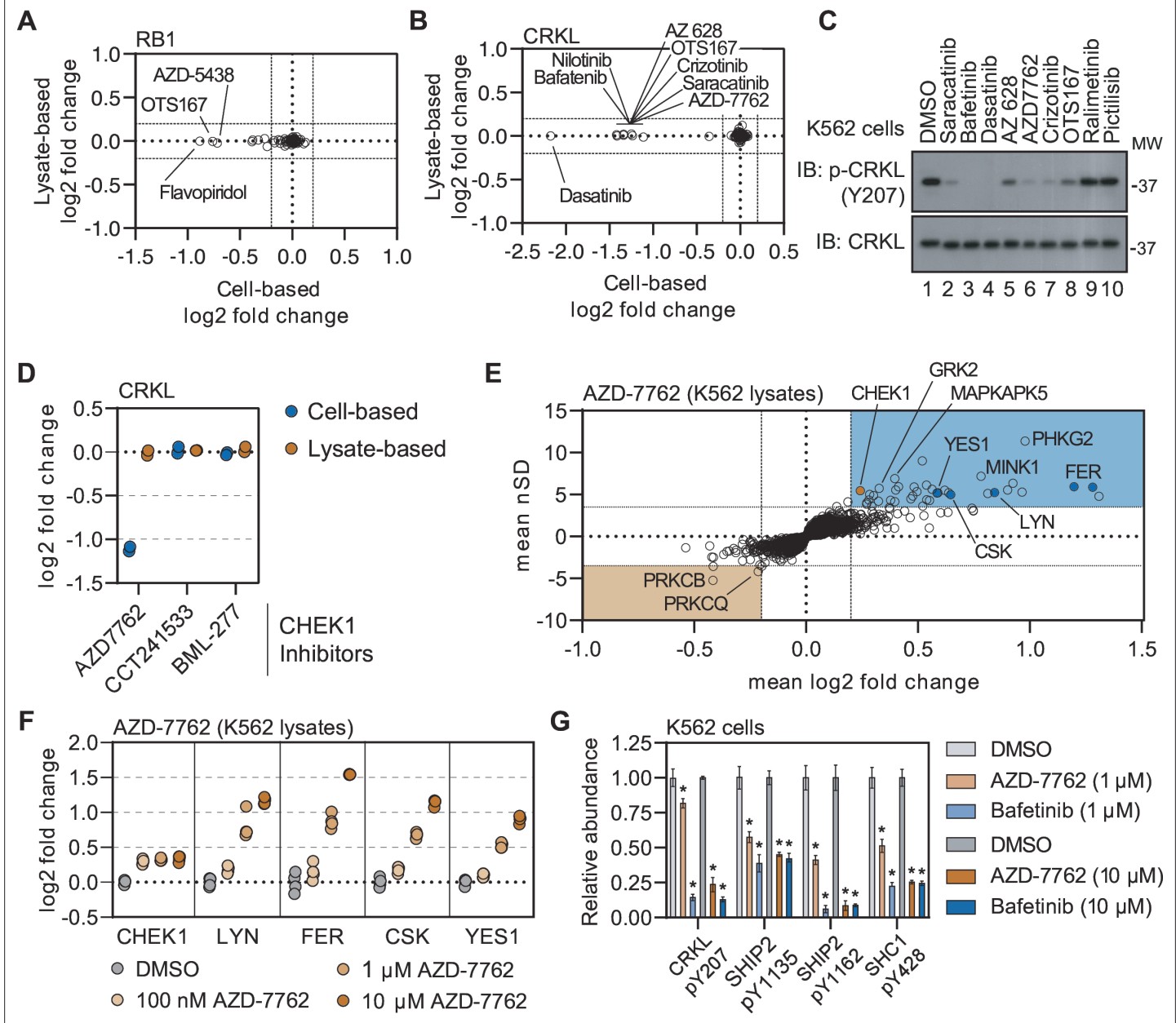

**Figure 5.** Disparities in cell- and lysate-based PISA can point toward secondary changes in protein thermal stability. (**A and B**) Plots depicting the mean log$_2$ fold change values for RB1 (**A**) and CRKL (**B**) in response to treatment with all compounds assayed using cell (**x**)- and lysate (**y**)-based PISA. Dashed lines indicate a log$_2$ fold change of +/-0.2. C. K562 cells were treated with each indicated compound at 10 μM for 15 min. Western blots were used to assess levels of p-CRKL (Y207) and total CRKL. D. A plot of the log2 fold change of CRKL in response to each indicated compound. (**E**) Plots highlighting the significant changes in protein solubility (thermal stability) following treatment with AZD-7762. The mean log$_2$ fold change of duplicate measurements is plotted on the x-axis and the mean nSD is plotted on the y-axis. Blue boxes contain proteins that exhibit an increase in solubility (increase in melting temperature) and orange boxes contain proteins that exhibit a decrease in solubility (decrease in melting temperature). Orange points represent the known target of AZD-7762 and the blue points represents tyrosine kinases that experience a significant change in soluble protein abundance (thermal stability). (**F**) K562 lysates were treated with the indicated concentrations of AZD-7762 for 15 min and assayed using PISA. Changes in protein thermal stability are represented as a log$_2$ fold change in soluble protein abundance for each treatment in reference to a DMSO-treated control. (**G**) K562 cells were treated with the indicated concentration of bafetinib or AZD-7762 for 15 min and assayed using phosphoproteomic profiling. Plots depict the relative abundance of several phosphorylation sites following treatment with the indicated compounds. * indicate significant changes, which were determined using a permutation-based FDR (FDR – 0.05, S0 – 0.1, N=4). Error bars represent the stabdard deviation.

The online version of this article includes the following source data and figure supplement(s) for figure 5:

**Source data 1.** PDF file containing original western blots for **Figure 5C**, indicating the relevant bands and treatments.

*Figure 5 continued on next page*

Figure 5 continued

**Source data 2.** Original files for western blot analysis displayed in *Figure 5C*.

**Source data 3.** This table contains lysate-based PISA data for K562 lysates treated with 100 nM, 1 µM, or 10 µM AZD-7762.

**Source data 4.** This table contains phosphoproteomic data for K562 cells treated with 1 µM AZD-7762, 10 µM AZD-7762, 1 µM bafetinib, or 10 µM bafetini.

**Source data 5.** This table contains proteomic data for K562 cells treated with 1 µM AZD-7762, 10 µM AZD-7762, 1 µM bafetinib, or 10 µM bafetinib.

**Figure supplement 1.** Disparities in cell- and lysate-based PISA can point toward secondary changes in protein thermal stability.

**Figure supplement 2.** Disparities in cell- and lysate-based PISA can point toward secondary changes in protein thermal stability.

**Figure supplement 3.** Disparities in cell- and lysate-based PISA can point toward secondary changes in protein thermal stability.

**Figure supplement 4.** Disparities in cell- and lysate-based PISA can point toward secondary changes in protein thermal stability.

number of significant changes in cells (102) vs. lysates (21). This disparity is driven by spliceosome subunits, which are significantly impacted in cell-based experiments but unaffected in lysates. This supports the interpretation that the apparent thermal destabilization of the spliceosome in cells is the result of secondary changes rather than direct ligand binding.

Exploiting the disparities between cell- and lysate-based data can be a powerful tool in determining compound mechanism of action in the absence of reliable evidence of target engagement. These studies were performed in K562 cells/lysates, a chronic myelogenous leukemia cell line that expresses the BCR-ABL fusion. Consistent with previous reports, treatment of K562 cells with BCR-ABL-targeted compounds did not induce any apparent change in the solubility (thermal stability) of their primary target (*Savitski et al., 2014*). We were, however, able to detect a change in solubility (thermal stability) for well-known BCR-ABL substrates—most notably CRKL and CRK (*Figure 5B* and *Figure 5—figure supplement 2A–D*). Importantly, the change in CRKL solubility only occurred in cell-based experiments and not in lysates (*Figure 5—figure supplement 2A and B*; *Savitski et al., 2014*). These data are consistent with a distal effector of thermal stability. A number of other compounds that largely target tyrosine kinases also caused a significant decrease in the solubility (thermal destabilization) of CRKL in K562 cells but not lysates (*Figure 5B*). We treated K562 cells with a small panel of these inhibitors and found that compounds that induce a decrease in the solubility of CRKL also inhibit phosphorylation at Y207 (*Figure 5C*). Therefore, the change in CRKL solubility correlates with a change in phosphorylation and is likely dependent on the inhibition of a primary compound target—such as BCR-ABL—that then inhibits CRKL phosphorylation. These data highlight that protein thermal stability measurements can reveal critical insights into primary and secondary effectors of compound mechanism of action, including kinase-substrate relationships (*Franken et al., 2015*).

## Using secondary changes to identify putative off-target effects

Previously, we demonstrated that the combined cell and lysate data could identify off-target inhibition through direct target engagement. Next, we wanted to determine if secondary effects could be used to determine putative off-target engagement in our screening data. In addition to tyrosine kinase inhibitors, we also observed a strong decrease in the solubility (thermal destabilization) of CRKL in response to treatment with the CHEK1 inhibitor AZD-7762 (*Figure 5B and C* and *Figure 5—figure supplement 2A*). Importantly, our screen contained two additional inhibitors known to engage CHEK1 (CCT241533 and BML-227), neither of which had any impact on CRKL solubility (*Figure 5D*). Therefore, the AZD-7762-dependent decrease in CRKL solubility is unlikely to be related to on-target inhibition of CHEK1 and, instead, likely stems from off-target engagement of other kinases. Consistent with our hypothesis, the lysate-based screening data revealed that AZD-7762 induced a change in the solubility of many tyrosine kinases including FER, LYN, CSK, and YES1 (*Figure 5E and F*). These putative interactors were not limited to tyrosine kinases but also included dozens of serine/threonine kinases.

In order to further corroborate the ability of AZD-7762 to engage and inhibit tyrosine kinases, we treated K562 lysates with an increasing concentration of this compound, ranging from 100 nM to 10 µM, and performed lysate-based PISA (*Figure 5—figure supplement 3A–C* and *Figure 5—source data 3*). Consistent with the screening data, we observed a dose-dependent increase in the solubility of FER, LYN, CSK, YES1, and other kinases (*Figure 5F* and *Figure 5—figure supplement 3A–C*). The annotated target of AZD-7762, CHEK1, on the other hand, experienced a similar change in solubility

in response to each dose of the compound, which implies that it is fully saturated even at the lowest dose (*Figure 5F*).

Previously, we showed that treatment of K562 cells with AZD-7762 prevented the phosphorylation of CRKL Y207 (*Figure 5C*). In order to further implicate the ability of AZD-7762 to inhibit tyrosine kinases in vivo, we treated K562 cells with 1 µM or 10 µM of AZD-772 or bafetinib (a known tyrosine kinase inhibitor employed in the primary screen) and performed phosphoproteomic profiling to quantify compound-dependent changes in phosphopeptide abundance (*Figure 5—figure supplement 4A–D* and *Figure 5—source data 4 and 5*). In addition to CRKL Y207, both bafetinib and AZD-7762 also caused a significant decrease in the abundance of other pY residues including SHIP2 Y1135 and Y1162 and SHC1 Y428 (*Figure 5G*). In addition to these specific pY sites, we also noticed that AZD-7762 caused considerably more total changes than bafetinib, which is consistent with AZD-7762 PISA profile and suggests that this compound is a highly promiscuous kinase inhibitor (*Figure 5—figure supplement 4A–F*).

## Library-scale assays connect consistent thermal stability changes

Binary comparisons of compounds with shared targets such as volasertib and BI 2536 (*Figure 1E*) revealed consistent thermal stability responses for known targets PLK1 and bromodomain proteins. Next, we set out to determine if small but consistent effects on the proteome due to drug treatment could reveal new information concerning compound engagement of proteins, protein classes, or protein complexes. To address this, we implemented a network-based approach using the correlation of solubility changes for each protein. To this end, we mapped the all-by-all correlation of proteins in the cell-based dataset. We filtered the 70,392,100 binary comparisons to include only the correlations in the top ~5% of absolute magnitude ($r_{spearman}$ >0.35). Because 70 of the 96 compounds used were kinase inhibitors, we further focused on correlated solubility changes within the 396 human kinases we quantified in our dataset (*Figure 6A*).

We first sought to determine if the network-based approach could reveal consistent thermal stability responses for kinases with conserved sequences or general kinase families (*Eid et al., 2017*). Strikingly, for the highly conserved 90 kDa ribosomal S6 kinases, we observed highly correlated solubility profiles for the three members, RPS6KA1, RPS6KA2, and RPS6KA3 (*Figure 6B*). We reasoned that this may be a function of the sequence conservation between these proteins or direct binding interactions (*Huttlin et al., 2021*; *Huttlin et al., 2017*). To this end we compared the cell and lysate data. In cells, RPS6KA1, RPS6KA2, and RPS6KA3 generally experienced a decrease in solubility following compound treatment. However, in lysates, these proteins generally experienced an increase in solubility, even by the same compound that decreased the solubility of proteins in cells, such as NVP-TAE-226. These data were in line with what we observed for sotrastaurin engagement of PRKCA (*Figure 4B*). Interestingly, within our thermal stability network, we observed that PRKCA solubility changes were significantly correlated with PRKCB and PRKCQ (*Figure 6—figure supplement 1A and B*).

Both RPS6K and PRKC subnetworks reveal correlation between kinases of the same kinome group. RPS6K proteins belong to the CAMK kinases, and PRKC proteins belong to the AGC kinases. To explore this within group correlation, we annotated the network of quantified human protein kinases based on their established groupings (AGC, Atypical, CAMK, CK1, CMGC, STE, TK, TKL, Other) along with a set of non-protein kinases (*Figure 6A*). Within the kinase network, we observed correlated solubility changes within and between families of kinases. For example, we observed correlated profiles between isoforms of p38α/MAPK14 (a CMGC kinase) and the MAPKAPK2/3 (CAMK kinases, *Figure 6C and D*). MAPKAPK2 and MAPKAPK3 both experienced an increase in solubility with treatment of the p38 kinase inhibitor doramapimod. Indeed, these proteins are known to both act as substrates of p38 and to form complexes in divergent human cancer cells (*Figure 6E*). We further observed interconnected correlation networks within larger kinase groups. The AGC kinases, including AKT1/2 and ROCK1/2 were particularly well correlated (*Figure 6F*). AKT1 and AKT2 was driven in part by their solubility change due to AKT inhibitor MK-2206 (*Figure 6G*).

Because protein kinases and non-protein kinases share similar structures and cofactor binding pockets, we reasoned that non-protein kinases may represent an important class of off-target effectors for our library compounds. Within the correlated thermal stability profile network, we observed strong correlation due to compound-based solubility changes within the non-protein kinase group. PIP4K2A,

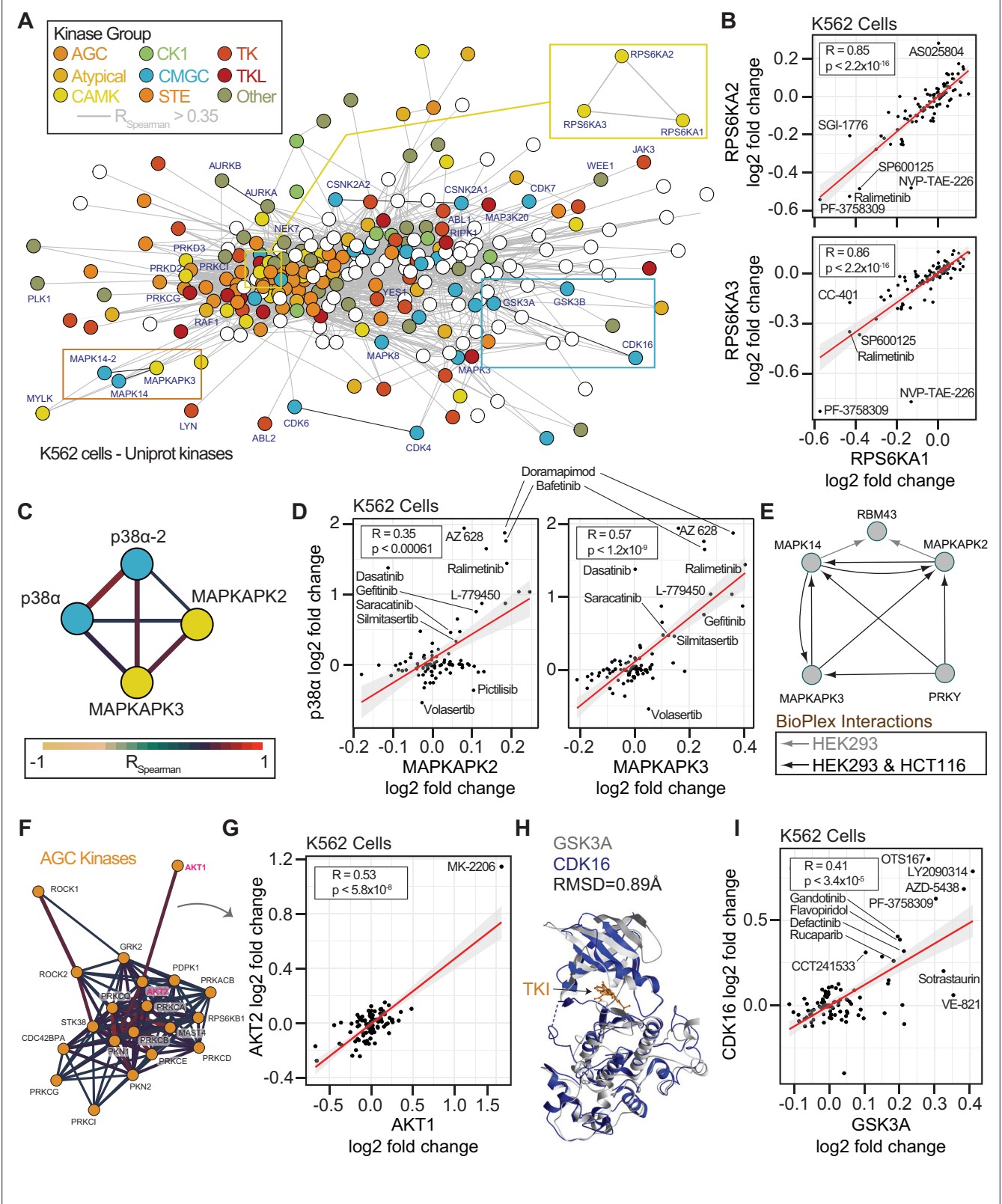

**Figure 6.** Concordance of protein engagement for protein complex members and structurally similar kinases. (**A**) Correlation network graph for all observed kinases in the cell-based PISA assays. Human kinases (nodes) are connected based on correlation coefficients ($R_{Spearman}$ >0.35). Nodes are colored based on kinase families. (**B**) Example of significant correlation between RPS6KA1, RPS6KA2, and RPS6KA3 with consistent but sometimes small changes in solubility (thermal stability). (**C**) Sub-graph of the network in A showing CMGC-family p38α/MAPK14 (isoforms 1 and 2) correlation with

*Figure 6 continued on next page*

*Figure 6 continued*

CAMK-family member kinases MAPKAPK2 and MAPKAPK3. Edges are colored based on $R_{Spearman}$. (**D**) Correlation plots for the subnetwork proteins in C. (**E**) Protein interaction network derived from the BioPlex interactome highlighting that MAPKAPK proteins directly interact with p38α/MAPK14 in multiple cell lines. (**F**) Sub-graph of the network in A highlighting the AGC protein kinases. Edges are colored based on $R_{Spearman}$. (**G**) AGC kinases AKT1 and AKT2 have a significant correlation driven in part by thermal stability effects of MK-2206. (H) Structural comparison of the CMGC kinases GSK3A (Alphafold) and CDK16 complexed with the TKI rebastinib (PDB: 5g6v). Alignment generated an RMSD of 0.89 Å. (**I**) Plot of the significant correlation between CDK16 and GSK3A derived from the network in A.

The online version of this article includes the following figure supplement(s) for figure 6:

**Figure supplement 1.** Concordance of protein engagement for protein complex members and structurally similar kinases.

PIP4K2B, and PIP4K2C solubilities were highly correlated in both cells and lysates (*Figure 6—figure supplement 1C–E*). For example, the PLK1 inhibitor BI 2536 caused an increase in the solubility of all three PIP4K proteins in both cells and lysates. It is worth noting that the magnitude of the solubility changes for these PIP4K proteins was higher in cells than in lysates consistent with loss of membranes and cellular metabolite concentrations (*Sridharan et al., 2019b*). Yet the thermal stability responses in both assays were highly consistent for all three proteins using either approach. We explored the observation of small but consistent changes further and observed significant correlation of the SRPK1, SRPK2, and RIOK2 kinases (*Figure 6—figure supplement 1F and G*). In our final datasets, we did not call any of these proteins as hits due to either the large numbers of solubility effects (SRPK1, SRPK2) or the small magnitude of changes (RIOK2). However, with correlations of 0.95 and 0.77 for SPRK1-SPRK2 and SPRK1-RIOK2, it is clear that these proteins respond consistently to chemical perturbation suggesting that our initial filters for full PISA datasets may mask some real and important protein-compound engagements.

Finally, we observed a strong correlation between the CMGC kinases CDK16, GSK3A, and GSK3B solubilities (*Figure 6A*). CDK inhibitors have been reported as active for GSK3 kinases owing to phylogenetic and structural conservation (*Figure 6H*; *Fischer, 2003*; *Meijer et al., 2003*). In our cell-based approach the inhibitor AZD-5438 has dual efficacy for GSK3A and CDK16 and we observed that the solubility responses for CDK16, GSK3A, and GSK3B were correlated (*Figure 6H*). This effect was driven in part by annotated inhibitors of CDKs (AZD-5438, flavopiridol) or GSKs (AZD-5438, LY2090314). Yet, known inhibitors of several kinases increased the solubility of both GSKs and CDK16, including MELK (OTSSP167), JAK (LY2784544), FAK (defactinib), and PAK4 (PF-3738309), as well as the PARP inhibitor rucaparib (*Figure 6I*). Interestingly, compounds such as VE821 (ATR), A205804 (ICAM-1), CC-401 (JNK), and L-779450 (Raf) exhibited GSK3 specific increases in solubility in cells. The consistent and differential responses of CDK16 and the GSKs offers clues into compound specificity or the utilization of combined effects in cells.

## Discussion

The determination of protein-compound engagement and mechanism of action is an essential aspect of the drug development process. Herein we present approachable methods for the implementation and interpretation of proteome-wide thermal shifts assays on at 96-well plate scale. Taking advantage of the eightfold increase in efficiency afforded by PISA, we catalogued compound-dependent changes in protein thermal stability in response to 96 compounds in living K562 cells and 70 compounds in native extracts. This study resulted in a total of 871,120 differential solubility (thermal stability) measurements in living cells and an additional 627,176 measurements in lysates. Exploiting this wealth of data, we were able to find clear evidence of on-target engagement for the majority of compounds tested in both cell- and lysate-based experiments. Furthermore, there were numerous potential examples of off-target engagement. In addition to direct target engagement, there were many secondary changes in protein thermal stability, as evidenced by changes in solubility that occurred independent of direct binding. Finally, using an all-by-all analysis of compounds and proteins, we demonstrated how structurally related proteins and protein complexes engage consistently with our library of compounds. Our study serves as an unbiased assessment of the current technology and serves as a comprehensive guide for ongoing work to enable library-scale study of protein-compound and protein-ligand interactions.

One of the most valuable features of the proteome-wide thermal shift assay, including PISA, is the ability to screen compounds in both living cells and native lysates (*Molina et al., 2013*; *Savitski et al., 2014*; *Franken et al., 2015*; *Huber et al., 2015*; *Becher et al., 2016*; *Becher et al., 2018*; *Dai et al., 2018*; *Gaetani et al., 2019*; *Sridharan et al., 2019b*; *Sabatier et al., 2022*; *Sabatier et al., 2021*). In this study, we further demonstrate that these two approaches are not redundant, but, rather, provide complimentary data that can be employed to help establish compound mechanism of action. First and foremost, each approach was similarly proficient in identifying compound-dependent changes of known targets (*Figures 1 and 3*). In both cell- and lysate-based PISA experiments, about 70% of all compounds (whose target was quantified) induced a significant change in the solubility of its well-established target(s). Furthermore, when the two datasets were combined, that figure increased to over 80%. This is because certain targets, like BRAF, for example were more prone to undergo a change in solubility in one experimental setting rather than the other (*Figure 4D*).

In general, a compound-dependent thermal shift that occurs in a lysate-based experiment is almost certain to stem from direct target engagement (*Savitski et al., 2014*; *Becher et al., 2016*). This is because cell signaling pathways and cellular structures are disrupted and diluted. Cell-based studies, on the other hand, have the added potential to identify the targets of pro-drugs that must be metabolized in the cell to become active and secondary changes that occur independent of direct engagement (*Savitski et al., 2014*; *Franken et al., 2015*; *Almqvist et al., 2016*; *Becher et al., 2016*; *Liang et al., 2022*). Consistent with this dichotomy, we found that, on average, compounds caused a greater number of thermal shifts (as evidenced by changes in solubility) in cells than lysates. This is further supported by the PF-3758309-dependent thermal destabilization of spliceosome subunits. Finally, we documented a number of examples of phosphorylation-dependent thermal shifts. For example, compounds targeting CDKs tended to impact the thermal stability of RB1. Likewise, compounds targeting tyrosine kinases induced a thermal destabilization of downstream phosphorylation targets like CRKL and CRK. In these examples, an inhibitor engages and inhibits a direct target, which, in turn, prevents the phosphorylation of a downstream target thereby inducing a change in thermal stability (*Savitski et al., 2014*).

The idea of area under the curve analysis (PISA) remains a relatively recent modification of the traditional proteome-wide thermal shift assay (*Gaetani et al., 2019*). Using this dataset, we were able to benchmark many aspects of this advance and highlight many important considerations when designing and interpreting these experiments. First, we highlight the importance of selecting an appropriate thermal window (*Li et al., 2020*). In this study we selected a temperature range of 48–58°C, however, it is possible that deploying multiple thermal windows could be a viable strategy for focused interrogation of a small number of high-value compounds. In utilizing thermal windows, the goal is to maximize the potential magnitudes of the final fold change measurements. Still, we observed a large range in the magnitudes of the $\log_2$ fold changes that were quantified for known compound targets. This is despite the fact, that we utilized a high treatment dose (10 µM) to maximize the magnitude of the fold change of the primary target. Ultimately, the data seemed to be consistent with previous studies that indicate the maximal change in thermal stability in protein specific (*Savitski et al., 2014*; *Becher et al., 2016*; *Sabatier et al., 2022*). Therefore, a $\log_2$ fold change of 2 for one compound-protein pair could be just as meaningful as a $\log_2$ fold change of 0.2 for another. In either case though we find that it is essential to optimize the thermal denaturation to ensure maximally useful effect sizes when comparing compounds.

While the primary screen was carried out at fixed dose, the increased throughput of PISA allowed for certain compounds to be assayed at multiple doses in a single experiment. In these instances, there was a clear dose-dependent change in thermal stability of primary targets, off-targets, and secondary targets. This not only helped corroborate observations from the primary screen, but also seemed to provide a qualitative assessment of relative compound potency in agreement with previous studies (*Savitski et al., 2014*; *Becher et al., 2016*; *Mateus et al., 2016*). Specifically, the compounds that most strongly impacted the thermal stability of targets, also acted as the most potent inhibitors. In order to be a candidate for this type of study, a target must have a large maximal thermal shift (magnitude of $\log_2$ fold change in solubility) because there must be a large enough dynamic range to clearly resolve different doses. From this study, PIK3CB (maximal $\log_2$ fold change ~0.2) would make a bad candidate, while p38α (maximal $\log_2$ fold change ~2.0) would be a good candidate. Importantly,

however, it should be possible to tune the thermal window in order to specifically increase the maximal $\log_2$ fold change of PI3KCB for dose-dependency studies.

Each of the compounds used in this study have a well-defined target and known mechanism of action. Furthermore, many of the compounds have been used extensively as tool compounds in basic research and as therapeutics in the treatment of multiple cancers. Despite the frequency with which many of these compounds have been used, the data generated in this study suggests that the current mechanistic understanding of many (if not all) of these compounds remains incomplete. For example, we found that the clinically deployed CDK4/6 inhibitor palbociclib has a dramatic impact on PLK1 thermal stability in live cells, is capable of inhibiting PLK1 activity in cell-based assays, and can be modelled into the PLK1 active site. Importantly, this activity could contribute to the mechanism of action of palbociclib and further supports the conclusions of a previous study that implicated palbociclib as a potential inhibitor of PLK1 (*Hafner et al., 2019*). Similarly, PISA helped reveal that the CHEK1 inhibitor AZD-7762 is, in fact, a highly promiscuous kinase inhibitor that, amongst many other kinases, appears to engage a number of tyrosine kinases and inhibit downstream signaling. While this compound has not been approved for clinical use, it has been used frequently as a tool compound to study the biological impact of CHEK1 inhibition. Additionally, the PLK1 inhibitor BI 2536 engages non-protein kinases such as PIP4K2A/B/C. As inhibition of PIP4K2s has been shown un hematological cells, this potentially suggests the mechanism of dose-limiting hematologic events upon patient treatment with BI 2536 (*Frost et al., 2012*; *Lima et al., 2022*). Overall, these examples, and the dataset as a whole, suggest that proteome-wide thermal shift assays and PISA, specifically, are valuable tools to help resolve the full range of compound activities. Despite this utility, however, it is important to note that these assays, alone, are unlikely to paint a full picture of mechanism of action. Instead, they represent a valuable tool that can be coupled with other MS-based and genetic approaches.

In the end, this study highlights the immense power of thermal stability assays in screening large collections of molecules to help define target engagement and mechanism of action on a proteome-wide scale. While this type of study is ideal for finding the targets of compounds that have been identified in high-throughput screening, they can provide valuable information in almost any context. This includes compounds that seemingly already have a well-defined target. Despite finding many ways to utilize this data, we expect that this dataset contains many interesting observations that can not only be used to better understand specific drugs, but also has the potential to reveal new biology. Moving forward, we hope that this data will serve as a valuable resource for those interested in drug development and the research community at large.

# Materials and methods

**Key resources table**

| Reagent type (species) or resource | Designation | Source or reference | Identifiers | Additional information |
|---|---|---|---|---|
| Antibody | Anti-CRKL (rabbit monoclonal antibody) | Cell Signaling Technologies | 38710 | 1:1000 |
| Antibody | Anti-p-CRKL Y207 (rabbit monoclonal antibody) | Cell Signaling Technologies | 3490 | 1:1000 |
| Antibody | Anti-RIPK1 (rabbit monoclonal antibody) | Cell Signaling Technologies | 3493 | 1:1000 |
| Antibody | Anti-p-RIPK1 S166 (rabbit monoclonal antibody) | Cell Signaling Technologies | 65746 | 1:1000 |
| Antibody | Anti-TCTP (rabbit monoclonal antibody) | Cell Signaling Technologies | 5128 | 1:1000 |
| Antibody | Anti-p-TCTP (rabbit monoclonal antibody) | Cell Signaling Technologies | 5251 | 1:1000 |
| Antibody | Goat anti-rabbit IgG-HRP secondary antibody | Santa Cruz Biotechnology | sc-2004 | 1:10000 |
| Cell line (*H. sapiens*) | K562 | ATCC | CLL-243 | |
| Cell line (*H. sapiens*) | HCT116 | ATCC | CLL-247 | |
| Chemical compound | AZD1152 | MedChemExpress | HY-10127 | |
| Chemical compound | Rosiglitazone | MedChemExpress | HY-17386 | |

*Continued on next page*

*Continued*

| Reagent type (species) or resource | Designation | Source or reference | Identifiers | Additional information |
|---|---|---|---|---|
| Chemical compound | Bafetinib | MedChemExpress | HY-50868 | |
| Chemical compound | AS-605240 | MedChemExpress | HY-10109 | |
| Chemical compound | MP7 | MedChemExpress | HY-14440 | |
| Chemical compound | Ibrutinib | MedChemExpress | HY-10997 | |
| Chemical compound | Gefitinib | MedChemExpress | HY-50895 | |
| Chemical compound | AZD-5438 | MedChemExpress | HY-10012 | |
| Chemical compound | Nilotinib | MedChemExpress | HY-10159 | |
| Chemical compound | GSK429286A | MedChemExpress | HY-11000 | |
| Chemical compound | PF-3758309 | MedChemExpress | HY-13007 | |
| Chemical compound | GSK126 | MedChemExpress | HY-13470 | |
| Chemical compound | OSI-027 | MedChemExpress | HY-10423 | |
| Chemical compound | Mubritinib | MedChemExpress | HY-13501 | |
| Chemical compound | Alisertib | MedChemExpress | HY-10971 | |
| Chemical compound | MK-2206 (dihydrochloride) | MedChemExpress | HY-10358 | |
| Chemical compound | Silmitasertib | MedChemExpress | HY-50855 | |
| Chemical compound | GSK343 | MedChemExpress | HY-13500 | |
| Chemical compound | NVP-TAE 226 | MedChemExpress | HY-13203 | |
| Chemical compound | BS-181 | MedChemExpress | HY-13266 | |
| Chemical compound | Erlotinib | MedChemExpress | HY-50896 | |
| Chemical compound | BML-277 | MedChemExpress | HY-13946 | |
| Chemical compound | BMS-911543 | MedChemExpress | HY-15270 | |
| Chemical compound | Givinostat | Cayman | 11045 | |
| Chemical compound | LY2090314 | MedChemExpress | HY-16294 | |
| Chemical compound | SD-208 | MedChemExpress | HY-13227 | |
| Chemical compound | AZ20 | MedChemExpress | HY-15557 | |
| Chemical compound | Epoxomicin | MedChemExpress | HY-13821 | |
| Chemical compound | Nexturastat A | MedChemExpress | HY-16699 | |
| Chemical compound | TAK-285 | MedChemExpress | HY-15196 | |
| Chemical compound | Sotrastaurin | MedChemExpress | HY-10343 | |
| Chemical compound | AZ 628 | MedChemExpress | HY-11004 | |
| Chemical compound | VE-821 | MedChemExpress | HY-14731 | |
| Chemical compound | Entinostat | MedChemExpress | HY-12163 | |
| Chemical compound | Filanesib | MedChemExpress | HY-15187 | |
| Chemical compound | AZD-7762 | MedChemExpress | HY-10992 | |
| Chemical compound | Crizotinib | MedChemExpress | HY-50878 | |
| Chemical compound | PCI-34051 | MedChemExpress | HY-15224 | |
| Chemical compound | Pinometostat | MedChemExpress | HY-15593 | |
| Chemical compound | SB 525334 | MedChemExpress | HY-12043 | |

*Continued on next page*

*Continued*

| Reagent type (species) or resource | Designation | Source or reference | Identifiers | Additional information |
|---|---|---|---|---|
| Chemical compound | IRAK-1–4 Inhibitor I | MedChemExpress | HY-13329 | |
| Chemical compound | Doramapimod | MedChemExpress | HY-10320 | |
| Chemical compound | JNJ-38877605 | MedChemExpress | HY-50683 | |
| Chemical compound | ZM 336372 | MedChemExpress | HY-13343 | |
| Chemical compound | Ispinesib | MedChemExpress | HY-50759 | |
| Chemical compound | MG-132 | MedChemExpress | HY-13259 | |
| Chemical compound | L-779450 | MedChemExpress | HY-12787 | |
| Chemical compound | Ralimetinib | Cayman | 23259 | |
| Chemical compound | Pictilisib | MedChemExpress | HY-50094 | |
| Chemical compound | GSK2656157 | MedChemExpress | HY-13820 | |
| Chemical compound | GSK-1070916 | MedChemExpress | HY-70044 | |
| Chemical compound | SGI-1776 | MedChemExpress | HY-13287 | |
| Chemical compound | Gandotinib | MedChemExpress | HY-13034 | |
| Chemical compound | Vorinostat | MedChemExpress | HY-10221 | |
| Chemical compound | GSK2606414 | MedChemExpress | HY-18072 | |
| Chemical compound | Y-33075 (dihydrochloride) | MedChemExpress | HY-10069 | |
| Chemical compound | CPI-203 | MedChemExpress | HY-15846 | |
| Chemical compound | Zotarolimus | MedChemExpress | HY-12424 | |
| Chemical compound | GSK-LSD1 (dihydrochloride) | MedChemExpress | HY-100546A | |
| Chemical compound | Flavopiridol | MedChemExpress | HY-10005 | |
| Chemical compound | Duvelisib | MedChemExpress | HY-17044 | |
| Chemical compound | TEPP-46 | MedChemExpress | HY-18657 | |
| Chemical compound | Saracatinib | MedChemExpress | HY-10234 | |
| Chemical compound | Dasatinib | MedChemExpress | HY-10181 | |
| Chemical compound | GW 501516 | MedChemExpress | HY-10838 | |
| Chemical compound | GSK2334470 | MedChemExpress | HY-14981 | |
| Chemical compound | CCT128930 | MedChemExpress | HY-13260 | |
| Chemical compound | Tozasertib | MedChemExpress | HY-10161 | |
| Chemical compound | BX-912 | MedChemExpress | HY-11005 | |
| Chemical compound | (+)-JQ-1 | MedChemExpress | HY-13030 | |
| Chemical compound | Tubastatin A | MedChemExpress | HY-13271A | |
| Chemical compound | ER-27319 maleate | Tocris | 2471 | |
| Chemical compound | BI 2536 | MedChemExpress | HY-50698 | |
| Chemical compound | CC-401 (hydrochloride) | MedChemExpress | HY-13022 | |
| Chemical compound | Encorafenib | MedChemExpress | HY-15605 | |
| Chemical compound | Entospletinib | MedChemExpress | HY-15968 | |
| Chemical compound | Rucaparib (Camsylate) | MedChemExpress | HY-102003 | |
| Chemical compound | Citarinostat | Cayman | 26173 | |

*Continued on next page*

*Continued*

| Reagent type (species) or resource | Designation | Source or reference | Identifiers | Additional information |
|---|---|---|---|---|
| Chemical compound | IC-87114 | MedChemExpress | HY-10110 | |
| Chemical compound | Vemurafenib | MedChemExpress | HY-12057 | |
| Chemical compound | OSU-03012 | MedChemExpress | HY-10547 | |
| Chemical compound | Defactinib | MedChemExpress | HY-12289 | |
| Chemical compound | PJ34 (hydrochloride) | MedChemExpress | HY-13688 | |
| Chemical compound | FTI-277 (hydrochloride) | MedChemExpress | HY-15872A | |
| Chemical compound | Idasanutlin | MedChemExpress | HY-15676 | |
| Chemical compound | SP600125 | MedChemExpress | HY-12041 | |
| Chemical compound | PRT062607 (Hydrochloride) | MedChemExpress | HY-15323 | |
| Chemical compound | Ro-3306 | MedChemExpress | HY-12529 | |
| Chemical compound | OTSSP167 (hydrochloride) | MedChemExpress | HY-15512A | |
| Chemical compound | SP2509 | MedChemExpress | HY-12635 | |
| Chemical compound | Lapatinib | MedChemExpress | HY-50898 | |
| Chemical compound | PLX-4720 | MedChemExpress | HY-51424 | |
| Chemical compound | CCT241533 (hydrochloride) | MedChemExpress | HY-14715B | |
| Chemical compound | Volasertib | MedChemExpress | HY-12137 | |
| Chemical compound | Tipifarnib | MedChemExpress | HY-10502 | |
| Chemical compound | A205804 | Cayman | 21252 | |
| Chemical compound | Quisinostat | MedChemExpress | HY-15433 | |
| Chemical compound | Palbociclib | Cayman | 12673 | |
| Chemical compound | Abemaciclib | Cayman | 21560 | |
| Chemical compound | Ribociclib | Cayman | 17666 | |
| Chemical compound | BI 2536 | Cayman | 17385 | |
| Chemical compound | NVP-TAE 226 | Cayman | 17684 | |
| Chemical compound | Necrostatin-2 | Cayman | 11657 | |
| Chemical compound | GSK2606414 | Cayman | 17367 | |
| Chemical compound | AZD-5438 | Cayman | 21598 | |
| Chemical compound | Tozaserib | SYNKinase | SYN-1092-M001 | |
| Chemical compound | Bafetinib | Cayman | 19169 | |
| Chemical compound | AZD7762 | Cayman | 11491 | |
| Chemical compound | Nocodazole | Cayman | 13857 | |
| Chemical compound | Z-VAD(Ome)-FMK | Cayman | 14463 | |
| Chemical compound | Birinapant | Cayman | 19699 | |
| Recombinant protein | Human recombinant TNF-α | Cayman | 32020 | |
| Software | Prism | GraphPad | 10.0.0 | |
| Software | Perseus | maxquant.net/perseus | | |
| Software | R | https://www.r-project.org/ | | |

## Cell culture

K562 cells were cultured in RPMI-1640 medium supplemented with 10% fetal bovine serum. For cell-based PISA experiments, cells were grown to approximately 1x10$^6$ cells/mL and immediately used for the assay. For lysate-based experiments, cells were grown to approximately 1x10$^6$ cells/mL, washed with phosphate-buffered saline, and flash frozen in liquid nitrogen. Cell pellets were stored at –80 °C until ready for use.

HCT116 cell were cultured in Dulbecco's Modified Eagle's Medium supplemented with 10% fetal bovine serum and 1 X penicillin-streptomycin. Cells were grown until ~80% confluency, harvested by scraping, washed with phosphate-buffered saline, and flash frozen in liquid nitrogen. Cell pellets were stored at –80 °C until ready for use.

## SDS-PAGE and western blotting

K562 and HCT116 cell lysates were combined with Laemmli buffer and resolved on Novex WedgeWell 4–20% Tris-Glycine gels (Invitrogen). Gels were transferred to an Immuno-Blot PVDF membrane (Bio-Rad). Membranes were immunoblotted with antibodies against CRKL (Cell Signaling Technologies (CST), 38710), p-CRKL Y207 (CST, 3490), RIPK1 (CST, 3493), p-RIPK1 S116 (CST, 65746), TCTP (CST, 5128), and p-TCTP S46 (CST, 5251). Following primary antibody, membranes were incubated in goat anti-rabbit IgG-HRP secondary antibody (Santa Cruz Biotechnology, sc-2004).

## Cell cycle synchronization

HCT116 cells were grown to approximately 50% confluency. The media was removed and replaced with media containing 100 ng/mL nocodazole (Cayman, 13857). The cells were allowed to incubate for 20 hr. After 20 hor, each compound was added at the desired concentration. The cells were allowed to further incubate for 1 hr. After 1 hr, the cells were washed three times with PBS and lysed in RIPA buffer. 20 µg of each lysate was separated by SDS-PAGE and analyzed with immunoblot.

## Initiation of necroptosis

HCT116 cells were grown to approximately 50% confluency. The media was removed and replaced with media containing 20 µM Z-VAD(Ome)-FMK (Cayman, 14463), 100 nM Birinapant (Cayman, 19699), and 10 ng/mL soluble recombinant human TNF-α (Cayman, 32020) (*Laurien et al., 2020*). Cells were incubated for 6 hr in the presence of various inhibitors or vehicle (DMSO). In the end, cells were washed three times with PBS and lysed in RIPA buffer. 20 µg of each lysate was separated by SDS-PAGE and analyzed with immunoblot.

## Cell-based PISA

K562 cells were grown to a concentration of 1x10$^6$ cells/mL. The cells were pelleted and resuspended on a 1:1 mixture of conditioned media and fresh media for a final concentration of 6x10$^6$ cells/mL. Each compound was added to fresh media at a 3 X concentration (30 µM). In order to initiate the experiment, 500 µL of cell suspension was mixed with 1 mL of 3 X treatment media and plated in a 24-well untreated tissue culture plate to achieve a final cell concentration of 2x10$^6$ cells/mL and a compound concentration of 10 µM. The cells were allowed to incubate for 30 min. After incubation, an equal volume of each culture was transferred to 10 PCR tubes. The PCR tubes were heated across a thermal gradient ranging from 48°C to 58°C (or 37–62°C) for 3 min to induce thermal denaturation. The samples were allowed to cool to room temperature and then an equal volume from each PCR tube was pooled. The cells in each pooled sample were washed once with PBS and then an equal volume of extraction buffer (1 X PBS pH 7.4, 0.5% NP-40, protease inhibitors) was added to added to each pellet. Samples were incubated for 10 min at 4 °C on a roller. Extracted samples were spun at 21,000x *g* for 90 min to separate insoluble aggregates from soluble protein. An equal volume from each soluble fraction was collected and prepared for LC-MS/MS analysis.

## Crude extract PISA (ysate-based PISA)

Frozen K562 or HCT116 pellets were thawed on ice and resuspended in lysis buffer (1 X PBS pH 7.4, 1 mM MgCl$_2$, protease inhibitor). The proteomes were extracted using a dounce homogenizer (20 strokes). The extracts were spun at 300 x *g* for 3 min to remove any unbroken cells. The resulting crude extract was diluted to 2 mg/mL in lysis buffer. Each compound was added to lysis buffer at

a 2 X concentration (usually 20 µM). In order to initiate the experiment, an equal volume of crude extract and treatment buffer were combined, to achieve a final protein concentration of 1 mg/mL and compound concentration of 10 µM, and incubated for 30 min. After incubation, an equal volume of each sample was transferred to 10 PCR tubes. The PCR tubes were heated across a thermal gradient ranging from 48°C to 58°C for 3 min to induce thermal denaturation. An equal volume from each PCR tube was pooled. An equal volume of extraction buffer (1 X PBS pH 7.4, 1% NP-40, protease inhibitors) was added to added to each pooled sample to achieve a final NP-40 concentration of 0.5%. Samples were incubated for 10 min at 4 °C on a roller. Extracted samples were spun at 21,000 x *g* for 90 min to separate insoluble aggregates from soluble protein. An equal volume from each soluble fraction was collected and prepared for LC-MS/MS analysis.

## LC-MS sample preparation

Samples (20 µg protein) were diluted in prep buffer (400 mM EPPS pH 8.5, 1% SDS, 10 mM tris(2-carboxyethyl)phosphine hydrochloride) and incubated at room temperature for 10 min. Iodoacetimide was added to a final concentration of 10 mM to each sample and incubated for 25 min in the dark. Finally, DTT was added to each sample to a final concentration of 10 mM. A buffer exchange was carried out using a modified SP3 protocol (*Hughes et al., 2019*; *Hughes et al., 2014*). Briefly, ~250 µg of Cytiva SpeedBead Magnetic Carboxylate Modified Particles (65152105050250 and 4515210505250), mixed at a 1:1 ratio, were added to each sample. 100% ethanol was added to each sample to achieve a final ethanol concentration of at least 50%. Samples were incubated with gentle shaking for 15 min. Samples were washed three times with 80% ethanol. Protein was eluted from SP3 beads using 200 mM EPPS pH 8.5 containing trypsin (Thermo Fisher Scientific, 90305R20) and Lys-C (Wako, 129–02541). Samples were digested overnight at 37 °C with vigorous shaking. Acetonitrile was added to each sample to achieve a final concentration of ~33%. Each sample was labelled, in the presence of SP3 beads, with ~60 µg of TMTPro 16plex reagents (Thermo Fisher Scientific). Following confirmation of satisfactory labelling (>97%), excess TMT was quenched by addition of hydroxylamine to a final concentration of 0.3%. The full volume from each sample was pooled and acetonitrile was removed by vacuum centrifugation for 1 hr. The pooled sample was acidified using formic acid and peptides were de-salted using a Sep-Pak 50 mg tC18 cartridge (Waters). Peptides were eluted in 70% acetonitrile, 1% formic acid and dried by vacuum centrifugation. The peptides were resuspended in 10 mM ammonium bicarbonate pH 8, 5% acetonitrile and fractionated by basic pH reverse phase HPLC. In total 24 fractions were collected. The fractions were dried in a vacuum centrifuge, resuspended in 5% acetonitrile, 1% formic acid and desalted by stage-tip. Finally, peptides were eluted in, 70% acetonitrile, 1% formic acid, dried, resuspended in 5% acetonitrile, 5% formic acid, and analyzed by LC-MS/MS.

For analysis of phosphopeptides, samples (100 µg protein) were prepared and digested as described above. Following digestion, acetonitrile was added to each sample to achieve a final concentration of ~33%. Each sample was labelled, in the presence of SP3 beads, with ~300 µg of TMTPro 16plex reagents (Thermo Fisher Scientific). Following confirmation of satisfactory labelling (>97%), excess TMT was quenched by addition of hydroxylamine to a final concentration of 0.3%. The full volume from each sample was pooled and acetonitrile was removed by vacuum centrifugation for 1 hr. The pooled sample was acidified using trifluoroacetic acid (TFA) and peptides were de-salted using a Sep-Pak 200 mg tC18 cartridge (Waters). Peptides were eluted in 70% acetonitrile, 1% formic acid and dried by vacuum centrifugation. Phosphopeptides were enriched using a High Select Phosphopeptide Enrichment Kit (Thermo Fisher Scientific, A32992). Following elution, phosphopeptides were acidified with formic acid and dried by vacuum centrifugation. The flow through from the phosphopeptide enrichment columns, which was retained for total proteome analysis, was fractionated as described previously. The dried phosphopeptides were solubilized in 5% acetonitrile, 0.1% TFA, desalted by stage-tip, and analyzed by LC-MS/MS.

## Offline basic reversed phase fractionation

TMT labeled peptides were solubilized in 5% acetonitrile/10 mM ammonium bicarbonate, pH 8.0 and ~300 µg of TMT labeled peptides were separated by an Agilent 300 Extend C18 column (3.5 µm particles, 4.6 mm ID and 250 mm in length). An Agilent 1260 binary pump coupled with a photodiode array (PDA) detector (Thermo Fisher Scientific) was used to separate the peptides. A 45-min linear

gradient from 10% to 40% acetonitrile in 10 mM ammonium bicarbonate pH 8.0 (flow rate of 0.25 mL/min) separated the peptide mixtures into a total of 96 fractions (36 s). A total of 96 Fractions were consolidated into 24 samples in a checkerboard fashion and vacuum dried to completion.

## Mass spectrometry data acquisition

Total proteome data were collected on Orbitrap Eclipse mass spectrometer (ThermoFisher Scientific) coupled to a Proxeon EASY-nLC 1000 (or 1200) LC pump (ThermoFisher Scientific). Peptides were separated using a 90–120 minute gradient at 500–550 nL/min on a 30 cm column (i.d. 100 µm, Accucore, 2.6 µm, 150 Å) packed inhouse. High-field asymmetric-waveform ion mobility spectroscopy (FAIMS) was enabled during data acquisition with compensation voltages (CVs) set as −40 V, −60 V, and −80 V (*Schweppe et al., 2019*). MS1 data were collected using the Orbitrap (Resolution – 60,000; Scan range – 400–1600 Th; Automatic gain control (AGC) - 4×10$^5$; Normalized AGC target – 100%; maximum ion injection time – 50ms). Determined charge states between 2 and 6 were required for sequencing, and a 90 s dynamic exclusion window was used. Data dependent mode was set as cycle time (1 s). MS2 scans were collected in the orbitrap after high-energy collision dissociation (HCD) fragmentation (Resolution – 50,000; AGC target – 1×10$^5$; Normalized AGC target – 200%; Normalized collision energy – 36; Isolation window – 0.5 Th; Maximum ion injection time – 86ms).

Phosphorylation data were collected on Orbitrap Eclipse mass spectrometer (ThermoFisher Scientific) coupled to a Proxeon EASY-nLC 1000 (or 1200) LC pump (ThermoFisher Scientific). Peptides were separated using a 90–120 minute gradient at 500–550 nL/min on a 30 cm column (i.d. 100 µm, Accucore, 2.6 µm, 150 Å) packed inhouse. High-field asymmetric-waveform ion mobility spectroscopy (FAIMS) was enabled during data acquisition with compensation voltages (CVs) set as −40 V, −60 V, and −80 V for the first shot and −45 V and −75 V for the second shot. MS1 data were collected using the Orbitrap (resolution – 120,000; maximum injection time – 50ms; AGC target – 4×10$^5$). Determined charge states between 2 and 5 were required for sequencing, and a 120 second dynamic exclusion window was used. Data dependent mode was set as cycle time (1 second). MS2 scans were performed in the Orbitrap after HCD fragmentation (resolution – 50,000; isolation window – 0.5 Da; collision energy – 36%; maximum injection time – 250ms; AGC – 1.5×10$^5$; Normalized AGC target – 300%).

## Mass spectrometry data analysis

Raw files were first converted to mzXML, and monoisotopic peaks were assigned using Monocle (*Rad et al., 2021*). Database searching included all human entries from Uniprot (downloaded on February 25$^{th}$, 2020). The database was concatenated with one composed of all protein sequences in the reversed order (*Elias and Gygi, 2007*). Sequences of common contaminant proteins (e.g., trypsin, keratins, etc.) were appended as well. Searches were performed with Comet (*Eng et al., 2013*) using a 50 ppm precursor ion tolerance and 0.02 Da product ion tolerance. TMT on lysine residues and peptide N termini (+304.207 Da) and carbamidomethylation of cysteine residues (+57.0215 Da) were set as static modifications, while oxidation of methionine residues (+15.9949 Da) was set as a variable modification. For phosphorylation searches, a variable modification of 79.9663 was set for serine, threeonine, and tyrosine residues. Phosphorylation site localization was determined using AScorePro (*Gassaway et al., 2022*). Peptide-spectrum matches (PSMs) were adjusted to a 1% false discovery rate (FDR; *Elias and Gygi, 2007*) PSM filtering was performed using linear discriminant analysis (LDA) as described previously (*Huttlin et al., 2010*) while considering the following parameters: XCorr, ΔCn, missed cleavages, peptide length, charge state, and precursor mass accuracy. Each run was filtered separately. Protein-level FDR was subsequently estimated at a data set level. For each protein across all samples, the posterior probabilities reported by the LDA model for each peptide were multiplied to give a protein-level probability estimate. Using the Picked FDR method (*Savitski et al., 2015*) proteins were filtered to the target 1% FDR level. TMT reporter ion intensities were measured using a 0.003 Da window around the theoretical *m/z* of each reporter ion. Proteins were quantified by summing reporter ion counts across all matching PSMs. Reporter ion intensities were adjusted to correct for the isotopic impurities of the different TMT reagents according to manufacturer specifications. Peptides were filtered to exclude those with a summed signal-to-noise (SN) <160 across all TMT channels. To control for different total protein loading within a TMT experiment, the summed protein quantities of each channel were adjusted to be equal within the experiment. For phosphorylation

experiments, the normalization factors that were applied to the associated proteome we also applied the phosphoproteome.

### Protein engagement hit calling

Compound engagement was determined based on relative thermal stability to DMSO controls. Owing to the fact that the duplicate analyses used in the initial screen would result in common statistical tests (Welch's t-test) being underpowered, we used a combination of fold changes compared to DMSO and individual protein variance to call hits. First, relative thermal stability to the DMSO controls was determined. Second, for each protein across all cells or lysate assays, the number of standard deviations (nSD) away from the mean thermal stability measurement (z-score) for a given protein was quantified. Cutoffs for fold change and z-score were determined to limit the number of hits derived from DMSO-treated samples. We maintained the same cutoffs across both datasets. We considered proteins to engage a compound if both replicates of the compound treatments resulted when the thermal stability fold change compared to DMSO greater than an absolute value of $\log_2 0.2$ and an absolute z-score greater than 3.5. This resulted in a false hit rate ($n_{DMSO-Hits}$) of 1% across all lysate-based assays and 3% for all cell-based assays.

### Statistical analyses

Follow-up PISA data were analyzed using Perseus (*Tyanova et al., 2016*). Significant changes were determined using a permutation-based FDR with the following settings – FDR – 0.05, S0 – 0.1, and number of randomizations – 250. Individual fold change values were calculated in reference to the mean of the vehicle-treated samples.

Correlation analyses were run in R version 4.3.1 using rank-based Spearman's rho ($r_{spearman}$) to minimize the effects of thermal stability-based outliers. The list of binary comparisons was filtered to include the top 5% of highly correlated and anticorrelate protein pairs. Significant pairwise correlations for individual comparisons were determined using the psych package.

### Structural modeling

Protein structures and associated ligands were modeled and aligned using the ICMBrowser v3.9 (Molsoft) and displayed either using the either Pymol v2.5.1 (*Figure 2H–2I*) or ICMBrowser (*Figure 6H*). PDB files were downloaded from RSCB PDB for the following identifiers: 2rku, 5l2i, 5g6v.

## Acknowledgements

We would like to acknowledge the Gygi and Schweppe labs for advice and technical assistance concerning methods development, experimental implementation, and data interpretation. We would also like to acknowledge our funding sources: R35GM150919-01 (DKS), R01GM067945-20 (SPG), Andy Hill CARE Distinguished Researcher Award (DKS), Cancer Consortium New Investigator Award (P30 CA015704, DKS), the Pew Charitable Trusts (DKS). JGV is the Mark Foundation for Cancer Research Fellow of the Damon Runyon Cancer Research Institute (DRG-2359–19).

## Additional information

### Competing interests

Steven P Gygi: SPG is on the advisory board for ThermoFisher Scientific, Cedilla Therapeutics, Casma Therapeutics, Cell Signaling Technology and Frontier Medicines. Devin K Schweppe: DKS is a consultant and/or collaborator with ThermoFisher Scientific, AI Proteins, Genentech, and Matchpoint Therapeutics. The other authors declare that no competing interests exist.

### Funding

| Funder | Grant reference number | Author |
|---|---|---|
| Pew Charitable Trusts | | Devin K Schweppe |

| Funder | Grant reference number | Author |
|---|---|---|
| National Institute of General Medical Sciences | R35GM150919 | Devin K Schweppe |
| W. M. Keck Foundation | | Devin K Schweppe |
| Andy Hill CARE Fund | | Devin K Schweppe |
| National Institute of General Medical Sciences | R01GM067945 | Steven P Gygi |
| National Cancer Institute | P30 CA015704 | Devin K Schweppe |

The funders had no role in study design, data collection and interpretation, or the decision to submit the work for publication.

## Author contributions

Jonathan G Van Vranken, Conceptualization, Resources, Data curation, Formal analysis, Validation, Investigation, Visualization, Methodology, Writing – original draft, Project administration, Writing – review and editing; Jiaming Li, Data curation, Software, Formal analysis, Methodology; Julian Mintseris, Data curation, Formal analysis; Ting-Yu Wei, Investigation; Catherine M Sniezek, Validation, Investigation, Visualization; Meagan Gadzuk-Shea, Conceptualization, Investigation, Writing – review and editing; Steven P Gygi, Conceptualization, Supervision, Writing – original draft, Project administration, Writing – review and editing; Devin K Schweppe, Conceptualization, Resources, Data curation, Software, Formal analysis, Supervision, Funding acquisition, Investigation, Visualization, Methodology, Writing – original draft, Project administration, Writing – review and editing

## Author ORCIDs

Jonathan G Van Vranken ⓘ https://orcid.org/0000-0002-8931-852X
Steven P Gygi ⓘ https://orcid.org/0000-0001-7626-0034
Devin K Schweppe ⓘ https://orcid.org/0000-0002-3241-6276

Reviewer #1 (Public review): https://doi.org/10.7554/eLife.95595.3.sa1
Reviewer #3 (Public review): https://doi.org/10.7554/eLife.95595.3.sa2
Author response https://doi.org/10.7554/eLife.95595.3.sa3

---

# Additional files

## Supplementary files

• Supplementary file 1. Library compounds. A list of all compounds utilized in the cell- and lysate-based PISA screens.

• Supplementary file 2. Sample multiplexing layout. This table contains the TMTPro channel assignments for all samples generated in the cell- and lysate-based screens.

• Supplementary file 3. Cell-based PISA full dataset. This table contains all the data from the cell-based screen in an easy to graph format.

• Supplementary file 4. Lysate-based PISA full dataset. This table contains all the data from the lysate-based screen in an easy to graph format.

• MDAR checklist

## Data availability

The mass spectrometry data have been deposited to the ProteomeXchange Consortium with the data set identifier PXD048009 (initial submission) and PXD053138 (revised submission). Information regarding TMT channel assignments and plex layout can be found in *Supplementary file 2*.

The following datasets were generated:

| Author(s) | Year | Dataset title | Dataset URL | Database and Identifier |
|---|---|---|---|---|
| Van Vranken JG, Li J, Mintseris J, Wei T, Sniezek CM, Gadzuk-Shea M, Gygi SP, Schweppe DK | 2024 | Large-scale characterization of drug mechanism of action using proteome-wide thermal shift assays (Initial submission) | https://www.ebi.ac.uk/pride/archive/projects/PDX048009 | PRIDE, PDX048009 |
| Van Vranken JG, Li J, Mintseris J, Wei T, Sniezek CM, Gadzuk-Shea M, Gygi SP, Schweppe DK | 2024 | Large-scale characterization of drug mechanism of action using proteome-wide thermal shift assays (Revised submission) | https://www.ebi.ac.uk/pride/archive/projects/PDX053138 | PRIDE, PDX053138 |

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
