## [Editor Report · eLife Assessment]

The study provides a **valuable** showcase of a workflow to perform large-scale characterization of drug mechanisms of action using proteomics in which on-target and off-targets of 166 compounds using proteome solubility analysis in living cells and cell lysates were determined. The evidence supporting the claims of the authors is **solid**, however, the inclusion of more replicate experiments and more statistical rigor would have strengthened the study. This will be of broad interest to medicinal chemists, toxicologists, computational biologists and biochemists.

---

## [Referee Report · Reviewer #1 (Public review)]

This paper describes proteome solubility analysis (PISA) of 96 compounds in living cells and 70 compounds in cell lysates. A wealth of information related to on- and off-target engagement is uncovered. This work fits well the eLife profile, will be of interest to a large community of proteomics researchers, and thus is likely to be reasonably highly cited.

---

## [Referee Report · Reviewer #3 (Public review)]

Summary:

This work aims to demonstrate how recent advances in thermal stability assays can be utilised to screen chemical libraries and determine compound mechanism of action. Focusing on 96 compounds with known mechanisms of action, they use the PISA assay to measure changes in protein stability upon treatment with a high dose (10uM) in live K562 cells and whole cell lysates from K562 or HCT116. They intend this work to showcase a robust workflow which can serve as a roadmap for future studies.

Strengths:

The major strength of this study is the combination of live and whole cell lysates experiments. This allows the authors to compare the results from these two approaches to identify novel ligand-induced changes in thermal stability with greater confidence. More usefully, this also enables the authors to separate primary and secondary effects of the compounds within the live cell assay.

The study also benefits from the number of compounds tested within the same framework, which allows the authors to make direct comparisons between compounds.

These two strengths are combined when they compare between CHEK1 inhibitors and suggest that AZD-7762 likely induces secondary destabilisation of CRKL through off-target engagement with tyrosine kinases.

Weaknesses:

One of the stated benefits of PISA compared to the TPP in the original publication (Gaetani et al 2019) was that the reduced number of samples required allows more replicate experiments to be performed. Despite this, the authors of this study performed only duplicate experiments. They acknowledge this precludes use of frequentist statistical tests to identify significant changes in protein stability. Instead, they apply an 'empirically derived framework' in which they apply two thresholds to the fold change vs DMSO: absolute z-score (calculated from all compounds for a protein) > 3.5 and absolute log2 fold-change > 0.2. They state that the fold-change threshold was necessary to exclude non-specific interactors. While the thresholds appear relatively stringent, this approach will likely reduce the robustness of their findings in comparison to an experimental design incorporating more replicates. Firstly, the magnitude of the effect size should not be taken as a proxy for the importance of the effect. They acknowledge this and demonstrate it using their own data for PIK3CB and p38α inhibitors (Figure 2B-C). They have thus likely missed many small, but biological relevant changes in thermal stability due to the fold-change threshold. Secondly, this approach relies upon the fold-changes between DMSO and compound for each protein being comparable, despite them being drawn from samples spread across 16 TMT multiplexes. Each multiplex necessitates a separate MS run and the quantification of a distinct set of peptides, from which the protein-level abundances are estimated. Thus, it is unlikely the fold-changes for unaffected proteins are drawn from the same distribution, which is an unstated assumption of their thresholding approach. The authors could alleviate the second concern by demonstrating that there is very little or no batch effect across the TMT multiplexes. However, the first concern would remain. The limitations of their approach could have been avoided with more replicates and use of an appropriate statistical test. It would be helpful if the authors could clarify if any of the missed targets passed the z-score threshold but fell below the fold-change threshold.

The authors use a single, high, concentration of 10uM for all compounds. Given that many of the compounds may have low nM IC50s, this concentration could be orders of magnitude above the one at which they inhibit their target. This makes it difficult to assess the relevance of the off-target effects identified to clinical applications of the compounds or biological experiments. The authors acknowledge this and use ranges of concentrations for follow-up studies (e.g. Figure 2E-F). Nonetheless, this weakness is present for the vast bulk of the data presented.

Aims achieved, impact and utility:

The authors have achieved their main aim of presenting a workflow which serves to demonstrate the potential value of this approach. However, by using a single high dose of each compound and failing to adequately replicate their experiments and instead applying heuristic thresholds, they have limited the impact of their findings. Their results will be a useful resource for researchers wishing to explore potential off-target interactions and/or mechanisms of action for these 96 compounds but are expected to be superseded by more robust datasets in the near future. The most valuable aspect of the study is the demonstration that combining live cell and whole cell lysate PISA assays across multiple related compounds can help to elucidate the mechanisms of action.

---

## [Author Response]

The following is the authors’ response to the original reviews.

**Public Reviews:**

**Reviewer #1 (Public Review):**
Summary:This is an interesting and potentially important paper, which however has some deficiencies.Strengths:A significant amount of potentially useful data.Weaknesses:One issue is a confusion of thermal stability with solubility. While thermal stability of a protein is a thermodynamic parameter that can be described by the Gibbs-Helmholtz equation, which relates the free energy difference between the folded and unfolded states as a function of temperature, as well as the entropy of unfolding. What is actually measured in PISA is a change in protein solubility, which is an empirical parameter affected by a great many variables, including the presence and concentration of other ambient proteins and other molecules. One might possibly argue that in TPP, where one measures the melting temperature change ∆Tm, thermal stability plays a decisive or at least an important role, but no such assertion can be made in PISA analysis that measures the solubility shift.

We completely agree with the insightful comment from the reviewer and we are very grateful that the point was raised. Our goal was to make this manuscript easily accessible to the entire scientific community, not just experts in the field. In an attempt to simplify the language, we likely also simplified the underlying physical principles that these assays exploit. In defense of our initial manuscript, we did state that PISA measures “a fold change in the abundance of soluble protein in a compound-treated sample vs. a vehicle-treated control after thermal denaturation and high-speed centrifugation.” Despite this attempt to accurately communicate the reviewer’s point, we seem to have not been sufficiently clear. Therefore, we tried to further elaborate on this point and made it clear that we are measuring differences in solubility and interpreting these differences as changes in thermal stability.

In the revised version of the manuscript, we elaborated significantly on our original explanation. The following excerpt appears in the introduction (p. 3):

“So, while CETSA and TPP measure a change in melting temperature (∆TM), PISA measures a change in solubility (∆SM). Critically, there is a strong correlation between ∆TM and ∆SM, which makes PISA a reliable, if still imperfect, surrogate for measuring direct changes in protein thermal stability (Gaetani et al., 2019; Li et al., 2020). Thus, in the context of PISA, a change in protein thermal stability (or a thermal shift) can be defined as a fold change in the abundance of soluble protein in a compoundtreated sample vs. a vehicle-treated control after thermal denaturation and high-speed centrifugation. Therefore, an increase in melting temperature, which one could determine using CETSA or TPP, will lead to an increase in the area under the curve and an increase in the soluble protein abundance relative to controls (positive log2 fold change). Conversely, a decrease in melting temperature will result in a decrease in the area under the curve and a decrease in the soluble protein abundance relative to controls (negative log2 fold change).”

And the following excerpt appears in the results section (p. 4):

“In a PISA experiment, a change in melting temperature or a thermal shift is approximated as a

significant deviation in soluble protein abundance following thermal melting and high-speed centrifugation. Throughout this manuscript, we will interpret these observed alterations in solubility as changes in protein thermal stability. Most commonly this is manifested as a log2 fold change comparing the soluble protein abundance of a compound treated sample to a vehicle-treated control (Figure 1 – figure supplement 1A).”

We have now drawn a clear distinction between what we were actually measuring (changes in solubility) and how we were interpreting these changes (as thermal shifts). We trust that the Reviewer will agree with this point, as they rightly claim that many of the observations presented in our work, which measures thermal stability, indirectly, are consistent with previous studies that measured thermal stability, directly. Again, we thank the reviewer for raising the point and feel that these changes have significantly improved the manuscript.

Another important issue is that the authors claim to have discovered for the first time a number of effects well described in prior literature, sometimes a decade ago. For instance, they marvel at the differences between the solubility changes observed in lysate versus intact cells, while this difference has been investigated in a number of prior studies. No reference to these studies is given during the relevant discussion.

We thank the reviewer for raising this point. Our aim with this paper was to test the proficiency of this assay in high-throughput screening-type applications. We considered these observations as validation of our workflow, but admit that our choice of wording was not always appropriate and that we should have included more references to previous work. It was certainly never our intention to take credit for these discoveries. Therefore, we were more than happy to include more references in the revised version. We think that this makes the paper considerably better and will help readers better understand the context of our study.

The validity of statistical analysis raises concern. In fact, no calculation of statistical power is provided.As only two replicates were used in most cases, the statistical power must have been pretty limited. Also, there seems to be an absence of the multiple-hypothesis correction.

We agree with the reviewer that a classical comparison using a t-test would be underpowered comparing all log2 normalized fold changes. We know from the data and our validation experiments that stability changes that generate log2 fold changes of 0.2 are indicative of compound engagement. When we use 0.2 to calculate power for a standard two-sample t-test with duplicates, we estimated this to have a power of 19.1%. Importantly, increasing this to n=3 resulted in a power estimate of only 39.9%, which would canonically still be considered to be underpowered. Thus, it is important to note that we instead use the distribution of all measurements for a single protein across all compound treatments to calculate standard deviations (nSD) as presented in this work. Thus, rather than a 2-by-2 comparison, we are comparing two duplicate compound treatments to 94 other compound treatments and 18 DMSO vehicle controls. Moreover, we are using this larger sample set to estimate the sampling distribution. Estimating this with a standard z-test would result in a p-value estimate <<< 0.0001 using the population standard deviation. Additionally, rather than estimate an FDR using say a BenjaminiHochberg correction, we estimated an empirical FDR for target calls based on applying the same cutoffs to our DMSO controls and measuring the proportion of hits called in control samples at each set of thresholds. Finally, we note that several other PISA-based methods have used fold-change thresholds similar to, or less than, those employed in this work (PMID: 35506705, 36377428, 34878405, 38293219).

Also, the authors forgot that whatever results PISA produces, even at high statistical significance, represent just a prediction that needs to be validated by orthogonal means. In the absolute majority of cases such validation is missing.

We appreciate this point and we can assure the reviewer that this point was not lost on us. To this point, we state throughout the paper that the primary purpose of this paper was to execute a chemical screen. Furthermore, we do not claim to present a definitive list of protein targets for each compound. Instead, our intention is to provide a framework for performing PISA studies at scale. In total, we quantified thousands of changes and feel that it would be unreasonable to validate the majority of these cases. Instead, as has been done for CETSA (PMID: 34265272), PISA (PMID: 31545609), and TPP (PMID: 25278616) experiments before, we chose to highlight a few examples and provide a reasonable amount of validation for these specific observations. In Figure 2, we show that two screening compounds—palbociclib and NVP-TAE-226—have a similar impact on PLK1 solubility as the two know PLK1 inhibitors. We then assay each of these compounds, alongside BI 2536, and show that the same compounds that impact the solubility of PLK1, also inhibit its activity in cell-based assays. Finally, we model the structure of palbociclib (which is highly similar to BI 2536) in the PLK1 active site. In Figure 4, we show that AZD-5438 causes a change in solubility of RIPK1 in cell- and lysate-based assays to a similar extent as other compounds known to engage RIPK1. We then test these compounds in cellbased assays and show that they are capable of inhibiting RIPK1 activity *in vivo*. Finally, in Figure 5, we show that treatment with tyrosine kinase inhibitors and AZD-7762 result in a decrease in the solubility of CRKL. We showed that these compounds, specifically, prevented the phosphorylation of CRKL at Y207. Next, we show that AZD-7762, impacts the thermal stability of tyrosine kinases in lysate-based PISA. Finally, we performed phosphoproteomic profiling of cells treated with bafetinib and AZD-7762 and find that the abundance of many pY sites is decreased after treatment with each compound. It is also worth stating that an important goal of this study was to determine the proficiency of these methods in identifying the targets of each compound. We do not feel that comprehensive validation of the “absolute majority of cases” would significantly improve this manuscript.

Finally, to be a community-useful resource the paper needs to provide the dataset with a user interface so that the users can data-mine on their own.

We agree and are working to develop an extensible resource for this. Owing to the size and complexities there, that work will need to be included in a follow-up manuscript. For now, we feel that the supplemental table we provide can be easily navigated the full dataset. Indeed, this has been the main resource that we have been emailed about since the preprint was first made public. We are glad that the Reviewer considers this dataset to be a highly valuable resource for the scientific community.

**Reviewer #2 (Public Review):**
Summary:Using K562 (Leukemia) cells as an experimental model, Van Vracken et. al. use Thermal Proteome Profiling (TPP) to investigate changes in protein stability after exposing either live cells or crude cell lysates to a library of anti-cancer drugs. This was a large-scale and highly ambitious study, involving thousands of hours of mass spectrometry instrument time. The authors used an innovative combination of TPP together with Proteome Integral Solubility Alternation (PISA) assays to reduce the amount of instrument time needed, without compromising on the amount of data obtained.The paper is very well written, the relevance of this work is immediately apparent, and the results are well-explained and easy to follow even for a non-expert. The figures are well-presented. The methods appear to be explained in sufficient detail to allow others to reproduce the work.

We thank the reviewer. One of our major goals was to make these assays and the resulting data approachable, especially for non-experts. We are glad that this turned out to be the case.

Strengths:Using CDK4/6 inhibitors, the authors observe strong changes in protein stability upon exposure to the drug. This is expected and shows their methodology is robust. Further, it adds confidence when the authors report changes in protein stability for drugs whose targets are not well-known. Many of the drugs used in this study - even those whose protein targets are already known - display numerous offtarget effects. Although many of these are not rigorously followed up in this current study, the authors rightly highlight this point as a focus for future work.Weaknesses:While the off-target effects of several drugs could've been more rigorously investigated, it is clear the authors have already put a tremendous amount of time and effort into this study. The authors have made their entire dataset available to the scientific community - this will be a valuable resource to others working in the fields of cancer biology/drug discovery.

We agree with the reviewer that there are more leads here that could be followed and we look forward to both exploring these in future work and seeing what the community does with these data.

**Reviewer #3 (Public Review):**
Summary:This work aims to demonstrate how recent advances in thermal stability assays can be utilised to screen chemical libraries and determine the compound mechanism of action. Focusing on 96 compounds with known mechanisms of action, they use the PISA assay to measure changes in protein stability upon treatment with a high dose (10uM) in live K562 cells and whole cell lysates from K562 or HCT116. They intend this work to showcase a robust workflow that can serve as a roadmap for future studies.Strengths:The major strength of this study is the combination of live and whole cell lysates experiments. This allows the authors to compare the results from these two approaches to identify novel ligand-induced changes in thermal stability with greater confidence. More usefully, this also enables the authors to separate the primary and secondary effects of the compounds within the live cell assay.The study also benefits from the number of compounds tested within the same framework, which allows the authors to make direct comparisons between compounds.These two strengths are combined when they compare CHEK1 inhibitors and suggest that AZD-7762 likely induces secondary destabilisation of CRKL through off-target engagement with tyrosine kinases.Weaknesses:One of the stated benefits of PISA compared to the TPP in the original publication (Gaetani et al 2019) was that the reduced number of samples required allows more replicate experiments to be performed. Despite this, the authors of this study performed only duplicate experiments. They acknowledge this precludes the use of frequentist statistical tests to identify significant changes in protein stability. Instead, they apply an 'empirically derived framework' in which they apply two thresholds to the fold change vs DMSO: absolute z-score (calculated from all compounds for a protein) > 3.5 and absolute log2 fold-change > 0.2. They state that the fold-change threshold was necessary to exclude nonspecific interactors. While the thresholds appear relatively stringent, this approach will likely reduce the robustness of their findings in comparison to an experimental design incorporating more replicates. Firstly, the magnitude of the effect size should not be taken as a proxy for the importance of the effect.They acknowledge this and demonstrate it using their data for PIK3CB and p38α inhibitors (Figures 2BC). They have thus likely missed many small, but biologically relevant changes in thermal stability due to the fold-change threshold. Secondly, this approach relies upon the fold-changes between DMSO and compound for each protein being comparable, despite them being drawn from samples spread across 16 TMT multiplexes. Each multiplex necessitates a separate MS run and the quantification of a distinct set of peptides, from which the protein-level abundances are estimated. Thus, it is unlikely the fold changes for unaffected proteins are drawn from the same distribution, which is an unstated assumption of their thresholding approach. The authors could alleviate the second concern by demonstrating that there is very little or no batch effect across the TMT multiplexes. However, the first concern would remain. The limitations of their approach could have been avoided with more replicates and the use of an appropriate statistical test. It would be helpful if the authors could clarify if any of the missed targets passed the z-score threshold but fell below the fold-change threshold.The authors use a single, high, concentration of 10uM for all compounds. Given that many of the compounds likely have low nM IC50s, this concentration will often be multiple orders of magnitude above the one at which they inhibit their target. This makes it difficult to assess the relevance of the offtarget effects identified to clinical applications of the compounds or biological experiments. The authors acknowledge this and use ranges of concentrations for follow-up studies (e.g. Figure 2E-F). Nonetheless, this weakness is present for the vast bulk of the data presented.

We agree that there is potential to drive off-target effects at such high-concentrations. However, we note that the concentration we employ is in the same range as previous PISA/CETSA/TPP studies. For example, 10 µM treatments were used in the initial descriptions of TPP (Savitski et al., 2014) and PISA (Gaetani et al., 2019). We also note that temperature may affect off-rates and binding interactions (PMID: 32946682) potentiating the need to use compound concentrations to overcome these effects.

Additionally, these compounds likely accumulate in human plasma/tissues at concentrations that far exceed the compound IC50 values. For example, in patients treated with a standard clinical dose of ribocicilb, the concentration of the compound in the plasma fluctuates between 1 µM and 10 µM. (Bao, X., Wu, J., Sanai, N., & Li, J. (2019). Determination of total and unbound ribociclib in human plasma and brain tumor tissues using liquid chromatography coupled with tandem mass spectrometry. *Journal of pharmaceutical and biomedical analysis*, *166*, 197–204. https://doi.org/10.1016/j.jpba.2019.01.017)

The authors claim that combining cell-based and lysate-based assays increases coverage (Figure 3F) is not supported by their data. The '% targets' presented in Figure 3F have a different denominator for each bar. As it stands, all 49 targets quantified in both assays which have a significant change in thermal stability may be significant in the cell-based assay. If so, the apparent increase in % targets when combining reflects only the subsetting of the data. To alleviate this lack of clarity, the authors could update Figure 3F so that all three bars present the % targets figure for just the 60 compounds present in both assays.

We spent much time debating the best way to present this data, so we are grateful for the feedback. Consistent with the Reviewer’s suggestion, we have included a figure that only considers the 60 compounds for which a target was quantified in both cell-based and lysate-based PISA (now Figure 3E). In addition, we included a pie chart that further illustrates our point (now Figure 3 – figure supplement 2A). Of the 60 compounds, there were 37 compounds that had a known target pass as a hit using both approaches, 6 compounds that had a known target pass as a hit in only cell-based experiments, and 6 compounds that had a known target pass as a hit in only lysate-based experiments.

Within the Venn diagram, we also included a few examples of compounds that fit into each category. Furthermore, we highlighted two examples of compound-target pairs that pass as a hit with one approach, but not the other (Figure 3 – figure supplement 2B,C). We would also like to refer the reviewer to Figure 4D, which indicates that BRAF inhibitors cause a significant change in BRAF thermal stability in lysates but not cells.

Aims achieved, impact and utility:The authors have achieved their main aim of presenting a workflow that serves to demonstrate the potential value of this approach. However, by using a single high dose of each compound and failing to adequately replicate their experiments and instead applying heuristic thresholds, they have limited the impact of their findings. Their results will be a useful resource for researchers wishing to explore potential off-target interactions and/or mechanisms of action for these 96 compounds, but are expected to be superseded by more robust datasets in the near future. The most valuable aspect of the study is the demonstration that combining live cell and whole cell lysate PISA assays across multiple related compounds can help to elucidate the mechanisms of action.
**Recommendations for the authors:**

**Reviewer #1 (Recommendations For The Authors):**
More specifically:P 1 l 20, we quantified 1.498 million thermal stability measurements.It's a staggering assertion, and it takes some reading to realize that the authors mean the total number of proteins identified and quantified in all experiments. But far from all of these proteins were quantified with enough precision to provide meaningful solubility shifts.

We can assure the reviewer that we were not trying to deceive the readers. We stated ‘1.498 million thermal stability measurements.’ We did not say 1.498 million compound-specific thermal stability shifts.’ We assume that most readers will appreciate that the overall quality of the measurements will be variable across the dataset, e.g., in any work that describes quantitation of thousands of proteins in a proteomics dataset. In accordance with the Reviewer’s suggestion, we have weakened this statement. The revised version of the manuscript now reads as follows (p. 1):

“Taking advantage of this advance, we quantified more than one million thermal stability measurements in response to multiple classes of therapeutic and tool compounds (96 compounds in living cells and 70 compounds in lysates).”

P 7 l 28. We observed a large range of thermal stability measurements for known compound-target pairs, from a four-fold reduction in protein stability to a four-fold increase in protein stability upon compound engagement (Figure 2A).PISA-derived solubility shift cannot be interpreted simply as a "four-fold reduction/increase in protein stability".

We thank the Reviewer for highlighting this specific passage and agree that it was worded poorly. As such, we have modified the manuscript to the following (p. 8):

“We observed a large range of thermal stability measurements for known compound-target pairs, from a four-fold reduction in protein solubility after thermal denaturation to a four-fold increase in protein solubility upon compound engagement (Figure 2A).”

P 8, l 6. Instead, we posit that maximum ligand-induced change in thermal stability is target-specific.Yes, that's right, but this has been shown in a number of prior studies.

We agree with the reviewer and accept that we made a mistake in how we worded this sentence, which we regret upon reflection. As such, we have modified this sentence to the following:

“Instead, our data appears to be consistent with the previous observation that the maximum ligandinduced change in thermal stability is target-specific (Savitski et al., 2014; Becher et al., 2016).”

P 11 l 7. Combining the two approaches allows for greater coverage of the cellular proteome and provides a better chance of observing the protein target for a compound of interest. In fact, the main difference is that in-cell PISA provides targets in cases when the compound is a pro-drug that needs to be metabolically processed before engaging the intended target. This has been shown in a number of prior studies, but not mentioned in this manuscript.

While our study was not focused on the issue of pro-drugs, this is an important point and we would be happy to re-iterate it in our manuscript. We thank the Reviewer for the suggestion and have modified the manuscript to reflect this point (p. 19):

“Cell-based studies, on the other hand, have the added potential to identify the targets of pro-drugs that must be metabolized in the cell to become active and secondary changes that occur independent of direct engagement (Savitski et al., 2014; Franken et al., 2015; Almqvist et al., 2016; Becher et al., 2016; Liang et al., 2022).”

While we are happy to make this change, we also would like to point out that the reviewer’s assertions that, “the main difference is that in-cell PISA provides targets in cases when the compound is a prodrug that needs to be metabolically processed before engaging the intended target” also may not fully capture the nuances of protein engagement effectors in the cellular context. Thus, we believe it is important to highlight the ability of cell-based assays to identify secondary changes in thermal stability.

P 11 l 28. These data suggest that the thermal destabilization observed in cell-based experiments might stem from a complex biophysical rearrangement. That's right because it is not about thermal stability, but about protein solubility which is much affected by the environment.

We agree that the readout of solubility is an important caveat for nearly every experiment in the family of assays associated with ‘thermal proteome profiling’. Inherently complex biophysical arrangements could affect the inherent stability and solubility of a protein or complex. Thus, we would be happy to make the following change consistent with the reviewer’s suggestion (p. 12):

“These data suggest that the decrease in solubility observed in cell-based experiments might stem from a complex biophysical rearrangement.”

P 12 l 7 (A). Thus, certain protein targets are more prone to thermal stability changes in one experimental setting compared to the other. Same thing - it's about solubility, not stability.

We thank the Reviewer for the recommendation and have modified the revised manuscript as follows (p. 13):

“Thus, certain protein targets were more prone to solubility (thermal stability) changes in one experimental setting compared to the other (Huber et al., 2015).”

P13 l 15. While the data suggests that cell- and lysate-based PISA are equally valuable in screening the proteome for evidence of target engagement... No, they are not equally valuable - cell-based PISA can provide targets of prodrugs, which lysate PISA cannot.

We have removed this sentence to avoid any confusion. We will not place any value judgments on the two approaches.

P 18 l 10. In general, a compound-dependent thermal shift that occurs in a lysate-based experiment is almost certain to stem from direct target engagement. That's true and has been known for a decade. Reference needed.

We recognize this oversight and would be happy to include references. The revised manuscript reads as follows:

“In general, a compound-dependent thermal shift that occurs in a lysate-based experiment is almost certain to stem from direct target engagement (Savitski et al., 2014; Becher et al., 2016). This is because cell signaling pathways and cellular structures are disrupted and diluted. Cell-based studies, on the other hand, have the added potential to identify the targets of pro-drugs that must be metabolized in the cell to become active and secondary changes that occur independent of direct engagement (Savitski et al., 2014; Franken et al., 2015; Almqvist et al., 2016; Becher et al., 2016; Liang et al., 2022).”

P 18 l 29. the data seemed to indicate that the maximal PISA fold change is protein-specific. Therefore, a log2 fold change of 2 for one compound-protein pair could be just as meaningful as a log2 fold change of 0.2 for another. This is also not new information.

We again appreciate the Reviewer for highlighting this oversight. The revised manuscript reads as follows:

“Ultimately, the data seemed to be consistent with previous studies that indicate the maximal change in thermal stability in protein specific (Savitski et al., 2014; Becher et al., 2016; Sabatier et al., 2022). Therefore, a log2 fold change of 2 for one compound-protein pair could be just as meaningful as a log2 fold change of 0.2 for another.”

P 19 l 5. Specifically, the compounds that most strongly impacted the thermal stability of targets, also acted as the most potent inhibitors. I wish this was true, but this is not always so. For instance, in Nat Meth 2019, 16, 894-901 it was postulated that large ∆Tm correspond to biologically most important sites ("hot spots") - the idea that was later challenged and largely discredited in subsequent studies.

Indeed, we agree with the Reviewer that there may be no essential connection between these. Rather, we are simply drawing conclusions from observations within the presented dataset.

Saying nothing about the work presented in the paper that the reviewer notes above, the referenced definition is also more nuanced “…we hypothesized that ‘hotspot’ modification sites identified in this screen (namely, those significantly shifted relative to the unmodified, bulk and even other phosphomodiforms of the same protein) may represent sites with disproportionate effects on protein structure and function under specific cellular conditions.” Indeed, in the response to that work, Potel et al. (https://doi.org/10.1038/s41592-021-01177-5) “agree with the premise of the Huang et al. study that phosphorylation sites that have a significant effect on protein thermal stability are more likely to be functionally relevant, for example, by modulating protein conformation, localization and protein interactions.”

Anecdotally, we also speculate that if we observe proteome engagement for two compounds (let’s say two ATP-competitive kinase inhibitors) that bind in the same pocket (let’s say the ATP binding site) and one causes a greater change in solubility, then it is reasonable to assume that it is a stronger evidence and we see evidence supporting this claim in Figure 2, Figure 3, Figure 4, and Figure 5.

It is also important to point out that previous work has also made similar points. This is highlighted in a review article by Mateus et al. (10.1186/s12953-017-0122-4). The authors state, “To obtain affinity estimates with TPP, a compound concentration range TPP (TPP-CCR) can be performed. In TPPCCR, cells are incubated with a range of concentrations of compound and heated to a single temperature.” In support of this claim, the authors reference two papers—Savitski et al., 2014 and Becher et al., 2016. We have updated this section in the revised manuscript (p. 20):

“While the primary screen was carried out at fixed dose, the increased throughput of PISA allowed for certain compounds to be assayed at multiple doses in a single experiment. In these instances, there was a clear dose-dependent change in thermal stability of primary targets, off-targets, and secondary targets. This not only helped corroborate observations from the primary screen, but also seemed to provide a qualitative assessment of relative compound potency in agreement with previous studies (Savitski et al., 2014; Becher et al., 2016; Mateus et al., 2017). Specifically, the compounds that most strongly impacted the thermal stability of targets, also acted as the most potent inhibitors. In order to be a candidate for this type of study, a target must have a large maximal thermal shift (magnitude of log2 fold change) because there must be a large enough dynamic range to clearly resolve different doses.”

Also, the compound efficacy is strongly dependent upon the residence time of the drug, which may or may not correlate with the PISA shift. Also important is the concentration at which target engagement occurs (Anal Chem 2022, 94, 15772-15780).

In our study, the time and concentration of treatment and was fixed for all compounds at 30 minutes and 10 µM, respectively. Therefore, we do not believe these parameters will affect our conclusions.

P 19 l 19. For example, we found that the clinically-deployed CDK4/6 inhibitor palbociclib is capable of directly engaging and inhibiting PLK1. This is a PISA-based prediction that needs to be validated by orthogonal means.

As we demonstrate in this work, the PISA assays serve as powerful screening methods, thus we agree that validation is important for these types of studies. To this end, we show the following:

• Proteomics: Palbociclib causes a decrease in solubility following thermal melting in cells.

• Chemical Informatic: Palbociclib is structurally similar to BI 2536.

• Protein informatics: Modeling of palbociclib in empirical structures of the PLK1 active site generates negligible steric clashes.

• Biochemical: Palbociclib inhibits PLK1 activity in cells.

We have changed this text to the following to clarify these points:

“For example, we found that the clinically-deployed CDK4/6 inhibitor palbociclib has a dramatic impact on PLK1 thermal stability in live cells, is capable of inhibiting PLK1 activity in cell-based assays, and can be modelled into the PLK1 active site.”

**Reviewer #2 (Recommendations For The Authors):**
I am wondering why the authors chose to use K562 (leukaemia) cells in this work as opposed to a different cancer cell line (HeLa? Panc1?). It would be helpful if the authors could present some rationale for this decision.

This is a great question. Two reasons really. First, they are commonly used in various fields of research, especially previous studies using proteome-wide thermal shift assays (PMID: 25278616, 32060372) and large scale chemical perturbations screens (PMID: 31806696). Second, they are a suspension line that makes executing the experiments easier because they do not need to be detached from a plate prior to thermal melting. We think this is a valuable point to make in the manuscript, such that non-experts understand this concept. We tried to communicate this succinctly in the revised manuscript, but would be happy to elaborate further if the Reviewer would like us to.

“To enable large-scale chemical perturbation screening, we first sought to establish a robust workflow for assessing protein thermal stability changes in living cells. We chose K562 cells, which grow in suspension, because they have been frequently used in similar studies and can easily be transferred from a culture flask to PCR tubes for thermal melting (Savitski et al., 2014; Jarzab et al., 2020).”

I note that integral membrane proteins are over-represented among targets for anti-cancer therapeutics. To what extent is the membrane proteome (plasma membrane in particular) identified in this work? After examining the methods, I would expect at least some integral membrane proteins to be identified. Do the authors observe any differences in the behaviour of water-soluble proteins versus integral membrane proteins in their assays? It would be helpful if the authors could comment on this in a potential revision.

We agree this is an important point when considering the usage of PISA and thermal stability assays in general for specific classes of therapeutics. To address this, we explored what effect the analysis of thermal stability/solubility had on the proportion of membrane proteins in our data (Author response image 1). Annotations were extracted from Uniprot based on each protein being assigned to the “plasma membrane” (07/2024). We quantified 1,448 (16.5% of total proteins) and 1,558 (17.3% of total proteins) membrane proteins in our cell and lysate PISA datasets, respectively. We also compared the proportion of annotated proteins in these datasets to a recent TMTpro dataset (Lin et al.; PMID: 38853901) and found that the PISA datasets recovered a slightly lower proportion of membrane proteins (~17% in PISA versus 18.9% in total proteome analysis). Yet, we note that we expect more membrane proteins in urea/SDS based lysis methods compared to 0.5% NP-40 extractions.

We were not able to find an appropriate place to insert this data into the manuscript, so we have left is here in the response. If the Reviewer feels strongly that this data should be included in the manuscript, we would be happy to include these data.

A final note: I commend the authors for making their full dataset publicly available upon submission to this journal. This data promises to be a very useful resource for those working in the field.

We thank the Reviewer for this and note that we are excited for this data to be of use to the community.

**Reviewer #3 (Recommendations For The Authors):**
There is no dataset PDX048009 in ProteomeXchange Consortium. I assume this is because it's under an embargo which needs to be released.

We can confirm that data was uploaded to ProteomeXchange.

MS data added to the manuscript during revisions was submitted to ProteomeXchange with the identifier – PDX053138.

Page 9 line 5 refers to 59 compounds quantified in both cell-based and lysate-based, but Figure 3E shows 60 compounds quantified in both. I believe these numbers should match.

We thank the Reviewer for catching this. In response to critiques from this Reviewer in the Public Review, we re-worked this section considerably. Please see the above critique/response for more details.

Page 10, lines 26-28: It would help the reader if some of the potential 'artefactual effects of lysatebased analyses' were described briefly.

We thank the Reviewer for raising this point. The truth is, that we are not exactly sure what is happening here, but we know that, at least, for vorinostat, this excess of changes in lysate-based PISA is consistent across experiments. We also do not see pervasive issues within the plexes containing these compounds. Therefore, we do not think this is due to a mistake or other experimental error. We hypothesize that the effect might result from a change in pH or other similar property that occurs upon addition of the molecule, though we note that we have previously seen that vorinostat can induce large numbers of solubility changes in a related solvent shift assays (doi: 10.7554/eLife.70784). We have modified the text to indicate that we do not fully understand the reason for the observation (p. 11):

“It is highly unlikely that these three molecules actively engage so many proteins and, therefore, the 2,176 hits in the lysate-based screen were likely affected in part by consistent, but artefactual effects of lysate-based analyses that we do not fully understand (Van Vranken et al., 2021).”

Page 24, lines 29-30 appear to contain a typo. I believe the '>' should be '<' or the 'exclude' should be 'retain'.

The Reviewer is completely correct. We appreciate the attention to detail. This mistake has been corrected in the revised manuscript.

Page 25, lines 5-7: The methods need to explain how the trimmed standard deviation is calculated.

We apologize for this oversight. To calculate the trimmed standard deviation, we used proteins that were measured in at least 30 conditions. For these, we then removed the top 5% of absolute log2 foldchanges (compared to DMSO controls) and calculated the standard deviation of the resulting set of log2 fold-changes. This is similar in concept to the utilization of “trimmed means” in proteomics data (https://doi.org/10.15252/msb.20145625), which helps to overcome issues due to extreme outliers in datasets. We have added the following statement to the methods to clarify this point (p. 27):

“Second, for each protein across all cells or lysate assays, the number of standard deviations away from the mean thermal stability measurement (z-score) for a given protein was quantified based on a trimmed standard deviation. Briefly, the trimmed standard deviation was calculated for proteins that were measured in at least 30 conditions. For these, we removed the top 5% of absolute log2 foldchanges (compared to DMSO controls) and calculated the standard deviation of the resulting set of log2 fold-changes.”

Page 25, lines 9-11 needs editing for clarity.

We tested empirical hit rates for estimation of mean and trimmed standard deviation (trimmedSD) thresholds to apply, to maximize sensitivity and minimizing the ‘False Hit Rate’, or the number of proteins in the DMSO control samples called as hits divided by the total number of proteins called as hits with a given threshold applied.

**Author response image 2. sa3fig2:** Hit calling threshold setting based on maximizing the total hits called and minimizing the False Hit Rate in cells (number of DMSO hits divided by the total number of hits).

**Author response image 3. sa3fig3:** Hit calling threshold setting based on maximizing the total hits called and minimizing the False Hit Rate in lysates (number of DMSO hits divided by the total number of hits).

Figure 1 supplementary 2a legend states: '32 DMSO controls'. Should that be 64?

We thank the Reviewer for catching our mistake. This has been corrected in the revised manuscript.

I suggest removing Figure 1 supplementary 3c which is superfluous as only the number it presents is already stated in the text (page 5, line 9).

We thank the Reviewer for the suggestion and agree that this panel is superfluous. It has been removed from the revised manuscript.

New data and tables added during revisions:

(1) Table 3 – All log2 fold change values for the cell-based screen. Using this table, proteincentric solubility profiles can be plotted (as in Figures 2D and others).

(2) Table 4 – All log2 fold change values for the lysate-based screen. Using this table, proteincentric solubility profiles can be plotted (as in Figures 2D and others).

(3) Figure 1 – Figure supplement 3H – Table highlighting proteins that pass log2 fold change cutoffs, but not nSD cutoffs and vice versa.

(4) Figure 2 – Panels H and I were updated with a new color scheme.

(5) Figure 3 – Updated main figure and supplement at the request of Reviewer 3.

• Figure 3E – Compares on-target hits for the cell- and lysate-based screens for all compounds for which a target was quantified in both screens.

• Figure 3 – Figure supplement 2 – Highlights on-target hits in both screens, exclusively in cells, and exclusively in lysates.

(6) Figure 5 – PISA data for K562 lysates treated with AZD-7762 at multiple concentrations.

• Figure 5F

• Figure 5 – Figure supplement 3A-C

• Figure 5 – Source data 2

(7) Figure 5 – Phosphoproteomic profiling of K562 cells treated with AZD7762 or Bafetinib.

• Figure 5G

• Figure 5 – Figure supplement 4A-F

• Figure 5 – Source data 3 (phosphoproteome)

• Figure 5 – Source data 4 (associated proteome data)